# A Systematic Review (1990–2021) of Wild Animals Infected with Zoonotic *Leishmania*

**DOI:** 10.3390/microorganisms9051101

**Published:** 2021-05-20

**Authors:** Iris Azami-Conesa, María Teresa Gómez-Muñoz, Rafael Alberto Martínez-Díaz

**Affiliations:** 1Department of Animal Health, Faculty of Veterinary Sciences, University Complutense of Madrid, 28040 Madrid, Spain; irisazami@ucm.es; 2Department of Preventive Medicine and Public Health, and Microbiology, Faculty of Medicine, University Autónoma of Madrid, 28029 Madrid, Spain; rafael.martinez@uam.es

**Keywords:** *Leishmania*, host, reservoir, wildlife, wild mammal, zoonoses, one health

## Abstract

Leishmaniasis are neglected diseases caused by several species of *Leishmania* that affect humans and many domestic and wild animals with a worldwide distribution. The objectives of this review are to identify wild animals naturally infected with zoonotic *Leishmania* species as well as the organs infected, methods employed for detection and percentage of infection. A literature search starting from 1990 was performed following the PRISMA methodology and 161 reports were included. One hundred and eighty-nine species from ten orders (i.e., Carnivora, Chiroptera, Cingulata, Didelphimorphia, Diprotodontia, Lagomorpha, Eulipotyphla, Pilosa, Primates and Rodentia) were reported to be infected, and a few animals were classified only at the genus level. An exhaustive list of species; diagnostic techniques, including PCR targets; infected organs; number of animals explored and percentage of positives are presented. *L. infantum* infection was described in 98 wild species and *L*. (*Viania*) spp. in 52 wild animals, while *L. mexicana*, *L. amazonensis*, *L. major* and *L. tropica* were described in fewer than 32 animals each. During the last decade, intense research revealed new hosts within Chiroptera and Lagomorpha. Carnivores and rodents were the most relevant hosts for *L. infantum* and *L*. (*Viannia*) spp., with some species showing lesions, although in most of the studies clinical signs were not reported.

## 1. Introduction

Included in the group known as neglected tropical diseases, the leishmaniases are a group of diseases caused by flagellated protozoan parasites from more than 20 species belonging to the genus *Leishmania*. The disease can appear with a variety of clinical pictures, depending on the species involved, the geographic region and the response of the host. Most people and animals infected by the parasite do not develop symptoms but, if present, the disease can follow three basic clinical forms in humans: cutaneous, mucocutaneous and visceral, while in animals, only cutaneous and/or visceral forms are observed. Leishmaniasis is a vector-borne disease transmitted by phlebotomine sand flies (order Diptera, family Psychodidae) with a worldwide distribution (Europe, Africa, the Americas, Asia, and one species recently described in Australia) and an endemic presence in more than 90 countries [1]. There are an estimated 350 million people at risk of *Leishmania* infection. The World Health Organization (WHO) estimates more than one million new cases per year in people around the world, making it one of the most relevant yet neglected parasitic diseases (https://www.who.int/leishmaniasis/en/, accesed on 8 January 2021). Leishmaniasis is one of the leading causes of morbidity in the world among infectious diseases and one of the leading causes of death among tropical diseases [2]. The disease is present in 98 of the 200 countries that collaborate with the WHO, and information is regularly updated by the organisation [3]. At least 39 described species of *Leishmania* can be found in the literature, in addition to a significant number of informal or incomplete citations [4,5]. Many authors consider that some of these species should be synonymous and that the *Leishmania* taxonomy should be revised and simplified [6]. A list with the zoonotic species of the genus *Leishmania* along with their main characteristics is shown in Table 1.

All species of the genus follow a biological cycle with the same pattern, alternating amastigote forms that replicate intracellularly in the vertebrate host, and promastigote forms that reproduce in the digestive system of the insect vector (Figure 1). Sand flies (mainly *Phlebotomus* and *Lutzomya* genera) become infected while feeding on a parasitised reservoir. Through the bite, they ingest macrophage-bearing blood and tissue with amastigotes. Natural vectors have been experimentally proven to be highly susceptible, and one or two parasites are sufficient to initiate an infection [7]. For a species of sand fly to be a vector of zoonotic *Leishmania,* it must meet five conditions: (1) be anthropophilic; (2) feed from reservoir hosts in cycles of zoonotic transmission; (3) be infected in nature with the same *Leishmania* species that infects humans; (4) harbour the complete development of the parasite until it becomes infective; (5) be capable of transmitting the parasite through the bite [8].

Other infectious routes, such as venereal and vertical transmission, have been proved for *L. infantum* in a canine host [9], and it is seriously considered in humans [10]. Biting is a route suspected for canids [11], and the oral route has been confirmed in hamsters [12], which has been suggested to be associated with the ingestion of micromammals by common genets and servals [13], or the ingestion of phlebotomines by insectivorous bats [14]. In another study, the presence of *Leishmania* amastigotes and promastigotes in the faeces of gorillas have been reported [15]. Indeed, translocation of bacteria from the gut to distant locations helped by dendritic cells is a phenomenon widely studied in the human microbiome nowadays [16,17].

A combination of strategies is required for the prevention and control of the disease, including early diagnosis and prompt and effective treatment, vector control, effective disease surveillance, control of animal reservoir hosts and social mobilisation and strengthening partnerships [18]. Domestic animals have been widely studied and, traditionally, dogs are considered the main animal reservoir, and cats and equines have been found in several studies infected with the parasite [19]. However, dogs were found with similar or even lower prevalence than wildlife during some human outbreaks, probably due to preventive measures applied [20,21]. For these reasons, the investigation of the role of wildlife in the infectivity and potential transmission of the parasite is an important step in order to control future outbreaks, and to monitor the endemicity of certain areas. A change in the factors influencing the abundance of vectors (i.e., deforestation, climatic change, and new urbanised areas), or the presence of potential animal reservoirs in a spatial and temporal coincidence with humans, are essential factors in the appearance of outbreaks.

According to the WHO [22], the term “reservoir” should be used only for animals that are sufficiently abundant and long-lived to be a food source for sand flies and that maintain intense contact with the sand fly vector in its environment. Several characteristics are necessary: (1) more than 20% of the specimens should be infected; (2) the course of infection should be long; (3) parasites should be available in the skin or blood in sufficient numbers to be taken up by the sand flies; (4) the parasites in the reservoirs and humans should be the same [23]. Reservoir hosts usually represent a large proportion of the mammalian biomass [23]. Roque and Jansen [24] applied the terms “maintenance hosts” for mammals than can be infected and maintain the infection and “amplifier hosts” for mammals that, besides maintaining the infection, may favour the transmission (for example with more parasites in the blood and skin for longer periods). In this context, many of the wild animals mentioned in this review could be considered as maintenance hosts that may serve as secondary reservoirs, if adequate conditions for disease dissemination are present, while only a few could be considered as amplifier hosts (i.e., animals proved to infect the vector or with high prevalence values and close proximity with humans).

This paper presents updated information on wildlife as potential reservoir hosts for all zoonotic *Leishmania* species, following a systematic review from 1990 to nowadays. Previous reviews should be examined for partial and prior information [19,23,24,25,26,27].

## 2. Methods

This systematic review was carried out following the Preferred Reporting Items for Systematic Reviews and Meta-Analyses (PRISMA) guidelines [28]. The main objective of this review was the identification of potential reservoirs of zoonotic species of *Leishmania*. Specific objectives were to (1) identify wild animal species naturally infected with zoonotic species of *Leishmania*; (2) recover information on the organs infected; (3) recover information on the techniques employed for detection and identification; (4) report data on the prevalence obtained in each study. These wild species should be investigated when an outbreak of leishmaniasis is present, or to monitor the endemicity of the disease in certain areas.

### 2.1. Search Strategy and Databases

Three databases were employed: PubMed (Medline), Scopus and Web of Science (WoS). The terms of search were “*Leishmania*” AND “wild” AND “host” OR “mammal”. The information was retrieved from 1990 to 27 February 2021 and included only articles in English and zoonotic species of *Leishmania* (Table 1). Each author revised one database and eliminated reports according to the exclusion criteria. Duplicates were removed from the list before the employment of Mendeley to upload the selected articles. After the screening, the articles selected for inclusion were deeply analysed according to the species of *Leishmania*: *Leishmania* (*Viannia*) spp. were analysed by MTGM, *Leishmania infantum* was analysed by IAC and the rest of the species were analysed by RAMD. When doubts arose, the three authors discussed them and reached a consensus. 

### 2.2. Exclusion Criteria

Automatic tools were employed to exclude some of the articles, while others were screened by the authors. Keywords for exclusion in Scopus were: animal experiment, animal model, mice, inbred C57BL, protozoan proteins, Bagg albino mouse, insect vectors, signal transduction, gene expression regulation, drug effect, inducible nitric oxide synthase, protein function, upregulation, wild type, macrophages, enzyme activity, mice knockout, cytokine production, Interleukin 10, Interleukin 4, mutant, mice inbred BALB C, unclassified drug, C57BL mouse, gene deletion, mutation, Th1 cell, cytokine, chemistry, Interleukin 12, protein expression, gamma interferon, arginase and CD4+T lymphocyte. Areas excluded in WoS were: research experimental medicine, virology, genetics heredity, biophysics, mycology, endocrinology metabolism, forestry, haematology, plant sciences, evolutionary biology, fisheries, oncology, physiology, polymer science and respiratory system.

Articles in Spanish, Portuguese, Turkish or French were excluded. Reviews, books and chapters of books, opinion articles, conference papers and letters were also excluded from the systematic review. Other criteria for exclusion were the following: experimental infections; clinical cases (except first citations); articles dealing with wild-type and genetically modified parasites; articles dealing only with human and/or domestic animals samples; articles dealing only with vectors or xenodiagnosis; articles dealing only with isolates obtained in previous studies; articles dealing only with phylogeny; studies with negative results to *Leishmania* infections; articles without enough information on the identification of *Leishmania* species; non-zoonotic species of *Leishmania*.

In total, 151 references were retrieved from the search of the databases, and ten more articles were found from other sources, and included (references from previous research articles and reviews) (Figure 2). All information contained in the tables was obtained from these articles, but additional references are included for the background. Recorded variables are included in the Appendix A: host, number of animals sampled, organs analysed, method of detection, prevalence and geographic area.

Due to the variability found among the studies and taking into account that wild animals are not easy to sample, a meta-analysis was not conducted. The main objective of this review was to update the list of potential reservoirs of the parasite and, for that reason, even studies with only one animal of a certain species were included.

## 3. Results

### 3.1. Result of the Search 

The database search identified 2018 records, 534 from Scopus, 935 from PubMed and 549 from WoS. After removal of articles by automation tools and by authors using title and abstract, duplicates were removed. Exclusion criteria were further applied and an outcome of 161 articles was reached: 151 articles retrieved from databases and ten from other sources (Figure 2).

### 3.2. Wild Animals Infected with Zoonotic Leishmania (Viannia) spp.

Among the species of *Leishmania* described in the Americas, *L. braziliensis* is one of the most widely investigated. *L. (Viannia) braziliensis* is the species more extensively distributed, and it has been described in Latin American countries, from Mexico to more southern countries. Data from wildlife include not only Brazil but also Venezuela, Colombia, Honduras, Belize, Peru, Panama and Argentina (Appendix A) [3,5]. Endemic leishmaniasis were present in several locations, such as the states of Sao Paulo and Minas Gerais in Brazil, especially where primary forest was substituted for human settlements [13]. Places such as coffee, banana or sugar cane plantations, ecotourism areas, or even chicken ruins and stables were known as breeding sites for the vector, and wild animals also became infected there [29,30,31]. Most of the studies were carried out in Brazil, especially in these endemic areas, whereas for other countries, such as Colombia, Peru, Bolivia, Venezuela or Argentina, infections were reported occasionally (Appendix A).

Blood and skin were the sites of detection in many of the published epidemiological works, since they are the most accessible for the hematophagous vector (Appendix A). However, when spleen, liver or bone marrow (BM) were included among the tissues analysed, prevalence rose, as the parasite tends to remain in these locations, even for species of *Leishmania* causing preferred cutaneous or mucocutaneous manifestations such as *L. (Viannia) braziliensis* or *L. mexicana*. Only one study employed oral swabs to successfully detect the DNA of the parasite [32].

Before the wide introduction of DNA amplification by PCR, fewer sensitive techniques were employed for the detection of *Leishmania*. Direct diagnosis of the parasite, such as examination of biopsies or imprints from skin or other organs, and culture in specific media were tested in rodents, marsupials and sloths in the past [24]. In fact, they are still being used in some studies today, although less frequently [30,31,33,34]. Serology has been extensively employed in domestic animals, but also in wildlife, as a sensitive and indirect evidence of *Leishmania* infection [30,35,36], and is the preferred method when studying the presence of *Leishmania* in wild carnivores [37] (Appendix A). Xenodiagnosis by exploring transmission of the vector was rarely employed [29], although this approach could prove the reservoir character of the hosts. Finally, experimental infections of wild and synanthropic rodents and monkeys was an approach not frequently used and will not be treated here, because it is beyond the scope of this review [24].

Since PCR became a routine technique to detect *Leishmania* in tissues, the list of infected hosts has lengthened. Brandao-Filho et al. [30] compared three diagnostic tests with spleen samples from 203 animals (rodents and marsupials) and found kDNA PCR over three times more sensitive (17.6%) than traditional methods such as microscopy of imprints (5.7%) or culture (1.3%). Serology seems to be less sensitive than molecular techniques, as it was shown for the prevalence of *Leishmania* in *Didelphis marsupialis* (8.1% by serology vs. 20% by PCR) [38] and when small mammals were analysed (5% serology vs. 8.8–23.2% by PCR) [39].

A vast number of primers, methods and targets for PCR detection can be found in the literature [5], but a few of them are widely employed in epidemiological studies in wildlife by several authors. Some targets are recognised as highly sensitive; this is the case for kinetoplast DNA (kDNA) which has more than 10,000 copies per cell [40]. PCR of kDNA has been employed for the detection of the parasite, followed by other approaches to determine the species of *Leishmania* present such as sequencing, RFLP or hybridisation (Appendix A). In some papers, the employment of PCR that amplify ITS1 or SSU allows also for the classification at the species level [5,14].

Animals infected with zoonotic species of *L. (Viannia)* belong to the orders Carnivora, Cingulata, Chiroptera, Didelphimorphia, Lagomorpha, Pilosa, Primata and Rodentia, this last order being the most extensively studied. 

Carnivores are usually blamed to be reservoirs of different species of *Leishmania*, and the scarce number of studies carried out revealed the presence of *Leishmania* (*Viannia*) spp. *L. braziliensis* DNAwas amplified by PCR in crab-eating dogs [34] and in Molina’s hog-nose skunks, this last species being pointed out as a reservoir by the authors, since a strain was isolated from one animal [41]. Specific antibodies employing serological techniques, such as direct agglutination test (DAT), were found in high percentages in hoary foxes, ring-tailed coatis and crab-eating racoons (50–100%), but the authors tested a small number of the animals for each species [37] (for details, see Appendix A).

The number of studies exploring the presence of *Leishmania* in bats has increased since 2013, when Shapiro et al. found DNA of the parasite in blood, liver and skin by PCR [42]. Since then, four more studies found *L.* (*Viannia*) spp. DNA in more than eight species of bats including hematophagous, insectivorous and frugivorous individuals [32,43,44,45].

Armadillos (order Cingulata) have been examined for *Leishmania* infection and found positive by PCR and culture plus zymodeme analysis in blood, spleen and liver [46] and in another study by PCR of the kDNA region, but the species of *Leishmania* present were not further investigated [34]. Regarding Lagomorpha, only one study demonstrated the presence of *L. braziliensis* in tapetis (*Sylvvilagus braziliensis*) in Colombia, employing xenodiagnosis and PCR followed by hybridisation [29] (Appendix A).

The presence of *L. braziliensis* has previously been reported in sloths, rodents and marsupials, and the reservoir character of these groups has been shown by several authors in the past [13,23]. The order Didelphimorphia, especially the white-eared opossum (*Didelphis albiventris*), was the focus of at least sixteen studies employing diverse PCRs, culture and serology, with highly variables percentages of infection [21,29,30,34,37,38,39,47,48,49,50,51,52,53,54,55] (Appendix A). In addition, xenodiagnosis was successful in this species as well as in the woolly-mouse opossum (*Micoureus demerarae*), which reinforces their role as main reservoirs of leishmaniasis [29,48]. One study found three two-toed-sloths (*Choloepus hoffmani*) infected with *L. panamensis* in Panama [56].

Rodents are the group most widely explored regarding *Leishmania* infections, both in natural and experimental conditions. The presence of *L. braziliensis* and other zoonotic species of the subgenus *Viannia* has been reported in 27 species including *Rattus rattus*, *Cerradomys subflavus, Necromys lasiurus*, *Nectomys squamipes* and *Mus musculus,* the latter being the species more often investigated [21,29,30,31,33,34,35,36,37,39,47,49,50,53,55,57,58,59,60,61,62,63] (Appendix A). This may be due to the fact of several reasons: their probable role as relevant reservoirs of leishmaniasis for humans, their proximity and high prevalence values (rats and domestic mouse), their abundance in ecological niches where phlebotomines reproduce, or to the successful attempts when xenodiagnosis or strain isolation were employed. Prevalence values varied from 1.2% to 100% depending on the sampling area, the sample size, the organs analysed, the diagnostic procedures employed for detection and characterisation and, probably, also on the age and lifespan of the sampled animals. Xenodiagnosis was successful in synanthropic species, such as black rats, and wild species, such as *Melanomys caliginosus* and *Mycroryzomys minutus.* The parasite was isolated by culture from *Akodon* spp., *Agouti paca, C. subflavus, N. lasiurus, R. rattus* and *Sigmodon hispidus*, but the PCR of kDNA was the preferred method for detection. When RFLP or sequencing was applied after PCR, the species could be determined, being *L. braziliensis* and *L. guyanensi* the most frequent. *L. naiffi*, *L. shawi* and *L. lainsoni* were obtained from rodents. Minor zoonotic species, such *L. shawi* and *L. naiffi,* were detected in species of the genus *Trichomys* [36], and *L. lainsoni* was mainly found in the big rodent paca (*Agouti paca*) [57] (Appendix A). Less frequent species of *Leishmania*, such as *L. peruviana,* were obtained from rodents and Didelphimorphia in Peru [47].

Only four studies were performed on primates employing PCR of KDNA, miniexon or ITS regions, some followed by sequencing or RFLP [34,37,64,65], while DAT was only employed in one study. *Leishmania* (*Viannia*) spp. was found between 8.6–100% of the animals analysed, with *L. braziliensis* being present, when identified to the species level [37].

All these investigations are summarized in Table 2.

In previous reviews [23,24], other species of wild animals were found infected with *Leishmania* (*Viannia*) spp., such as rodents (*Coendu* sp., *Rhipidomys leucodactylus*, *Heteromys dermarestianus*, *Proechymis semispinosus*, *Trichomys pachyurus*), sloths (*Choleopus didactylus*, *Bradypus infuscatus*, *Bradypus tridactylus*), anteanters (*Tamandua tetradactyla*), primates (*Aotus trivirgatus*, *Cebus apella*, *Chiropotes satanas*, *Sanguinus geoffroyi*) and carnivores (*Nasua nasua*, *Potos flavus*). For more details, see previous articles dealing with leishmaniasis in the Americas [23,24].

### 3.3. Wild Animals Infected with Leishmania amazonensis

*L. amazonensis* was described in countries from Central and South America, where data were available, including Costa Rica, Panama, Venezuela, Colombia, Ecuador, Peru, Argentina, Uruguay, French Guiana, Surinam, Brazil and Bolivia, the last two countries being the most widely studied [3,5]. This review includes updated information of wild animals infected with the parasite from surveys carried out in Brazil, Bolivia and Argentina.

Among carnivores, *L. amazonensis* was detected only in one of two skunks analysed from a focus of leishmaniasis in Bolivia [41]. More information was retrieved from three studies including several species of bats [14,45,66]. Most of the species analysed were insectivorous bats, but the parasite was also detected in hematophagous (*Desmodus rotundus*), nectarivorous and omnivorous species. Prevalence values varied from 1% to 25%, probably depending on the geographic area, the species and the organs or techniques employed for detection. Higher values were observed in *Sturnira lilium* and *Eumops auripendulus* from urban areas and remnants of primitive forest of Sao Paulo (Brazil), employing nested PCR from liver and spleen (Appendix A). 

Primates and opossum have been scarcely reported with *L. amazonensis* in the last 30 years, but the species was recently detected by PCR of the ITS region in the ear tissue of 2.8% of 209 black howler monkeys (*Alouatta caraya*) from Argentina [64]. In addition, a clinical case of a spider monkey (*Ateles paniscus*) from a zoo in Brazil, which showed weight loss and pale mucous membranes, was further confirmed by PCR and RFLP from blood [67]. *L amazonensis* DNA was also detected in 1.1% of the analysed woolly-mouse opossum (*Marmosa paraguayanus*) from Brazil [48] (Appendix A).

Three studies from Bolivia and Brazil reported the presence of *L. amazonensis* in blood or skin (tail or ear) by PCR followed by sequencing, in 7.1–33.3% of the analysed rodents (*Hylaeamys, Oryzomys, Akodon, Necromys* and *Olygoryzomis* genera) [35,41,68]. Some of these rodents displayed old lesions including scars on the tail or ear [68].

Animals found infected with *L. amazonensis*, as weel as the techniues employed, are summarized in Table 3.

Further information on other species infected with *L. amazonensis* can be found in previous reviews [23,24], and include rodents (*Dasyprocta* spp., *Oligoryzomis* spp., *Orzyomis melanotis*, *Proechymis* spp., *Trichomys apereoides*, *Sciurus vulgaris*), carnivores (*Cerdocyon thous*, *Potos flavus*), anteaters (*Tamandua tetradactyla*), marsupials (*Didelphis marsupialis*, *Metachirus nudicaudatus*, *Philander opossum* and *Marmosa cinerea*) and primates (*Saguinus geoffroyi*). 

### 3.4. Wild Animals Infected with Leishmania mexicana

In this review, data regarding infection with *L. mexicana* in wild animals were mainly from the USA and Mexico, two countries where the parasite is frequently reported, but data from rodents and carnivores from Brazil and Bolivia were also included (Appendix A). The species was present in other American territories such as Venezuela, Colombia, Ecuador and all countries in Central America [3,5].

According to the data from the systematic review, thirty-one species of wild animals from six orders were found to be infected with *L. mexicana*. In carnivores, only one out of two Molina’s hog-nosed skunks (*Conepatus chinga rex*) were infected with *L. mexicana* in Bolivia, and the parasite was isolated by inoculation in hamster and subsequently analysed by isoenzyme analysis and hybridisation [41]. Samples from seven grey foxes (*Urocyon cinereoargenteus)* were analysed by ELISA, showing 100% prevalence in Mexico [69] (Appendix A). Both species should be considered as sentinel or even reservoirs, due to the parasite’s isolation and high values of positivity.

Thirteen species of bats (order Chiroptera) were also found infected with the parasite in Mexico, employing skin, heart, liver and spleen in a PCR of kDNA and SSU [5]. The authors found infection rates ranging from 4–100% of the animals [70].

In rodents, eleven species were infected in ten different surveys, with *Handleyomis* (sin. *Oryzomis*) *melanotis*, *Ototylomis phyllotis*, *Peromyscus yucatanicus* and *Sigmodon hispidus* being the species with the higher levels of infection (100% in at least one study), which may indicate their role as reservoirs of the disease [33,58,71,72,73,74,75,76,77,78]. Within the order Didelphimorphia, the Mexican mouse opossum (*Marmosa mexicana*) [71] and the northern anteater [79] were found to be infected in Mexico employing PCR. 

The order Primates was less explored, and only indirect evidence of the infection was reported by serology (ELISA, IFAT and Western blot). A prevalence of 5–37.5% was found in two species of howler monkeys (*Alouatta palliate* and *A. pigra*) in Mexico [80].

The base of the tail was the election site for detection or isolation of *L. mexicana* in rodents and marsupials, with 100% of infection in many studies in which animals with lesions were sampled [71,72,73,77], but the ear and foot were also included [76] (Appendix A). When other organs were investigated, such as liver, spleen, kidney or heart, they were also infected, but at lower percentages (11–66.7%) [72]. Heart, liver, spleen and skin were also employed to find infections in Chiroptera [70], while lymph nodes, lung, spleen, liver and kidney tissues were used in the northern anteater, with DNA detection by PCR only in spleen [79]. 

In general, wild animals showed mild clinical signs of leishmaniasis and no external signs were reported in the orders Carnivora, Chiroptera, Pilosa and Primates. On the contrary, rodents and marsupials were reported with cutaneous clinical signs in most of the surveys, including swollen skin, depigmentation, ulcers, alopecia and erythema, mainly at the base of the tail. This fact can be explained because the authors were searching for lesions to find reservoirs of the disease [71,72,77]. 

Wild animals infected with *L. mexicana*, along with techniques and organs or tissues positive to the parasite are shown in Table 4.

In previous reviews, several species from publications prior to 1990 were reported to be infected with *L. mexicana* (*Agouti paca*, *Marmosa robinson*, *Nyctomys sumichrasti*, *Oryzomis capito*, *Proechymis* spp., *Reithrodontomys gracilis*) [23,24].

### 3.5. Wild Animals Infected with Leishmania infantum (L. chagasi)

*L. infantum* is the most globally distributed of all species of zoonotic *Leishmania*. Australia is considered free of *L. infantum*, but the protozoan is present on almost all continents with available data, including Southern Europe, Africa, Asia and the Americas from north (excluding Alaska and Canada) to south. African countries and Brazil report more than 90% of the human VL cases around the world, but detailed characterised focusses are more frequently reported in Brazil and Mediterranean countries (North Africa and South Europe) [3,5].

Techniques employed to detect infection with *L. infantum* in different parts of the world are similar to those previously described for *L. braziliensis*. Serology was mainly employed in carnivores, primates and occasionally in marsupials or other species, such as rodents or Lagomorpha (Appendix A), while the rest of the species were examined preferentially by PCR. Among the serological techniques, ELISA, IFAT, DAT or rapid test (rK39) were extensively employed. The most frequent target, again, was kDNA, but other targets, such as SSU and ITS1 and the repeat region, were also used in several studies and animal species. Less frequently used targets include cytochrome B (Cyt B), HSP70, ITS2, glyceraldehyde phosphate hydrogenase (GAPDH) and α-tubulin (for details, see Appendix A). Xenodiagnosis or culture were employed only in a few occasions.

Blood, skin, liver and spleen were the most employed tissues for PCR detection, but heart, lungs, lymph nodes, intestines, kidney and bone marrow were also used in several studies. Blood was more frequently employed in carnivores, marsupials and primates, because it is easier to obtain, while other tissues were accessible only during post-mortem examinations or after fatal clinical cases or euthanasia of the animals. This was the case for rodents, some bats, several clinical cases of carnivores, and road-killed mammals. Hair and eye swabs were also successfully tested in some studies (Appendix A).

#### 3.5.1. *L. infantum* in the Americas

According to the literature, eight orders of wild animals are infected with *L. infantum* in the Americas: Carnivores, Chiroptera, Cingulata, Didelphimorphia, Lagomorpha, Pilosa, Primates and Rodentia (Appendix A). Carnivores were the most widely studied, mainly because domestic and wild carnivores are considered reservoirs of the disease, but also because clinical cases are more frequently reported in them, both in nature and in zoological parks [24,25]. The crab-eating fox (*Cerdocyon thous*) is a widespread carnivore in South America that can act as a reservoir of leishmaniasis for humans, since it can be found in forest locations as well as in residential areas. It was found to be infected with *L. infantum* or exposed to the parasite (positive serology) in several publications, some of which were clinical cases, and thus were not considered in this study. The percentage of infection varied widely among the studies when including more than one animal (4–75%), and exposure to the parasite was demonstrated by serology (i.e., ELISA, IFAT), while culture, microscopy of smears, PCR followed by sequencing and inoculation of hamsters were employed to detect the parasite [81,82,83,84,85,86,87,88,89]. Several organs and tissues tested positive via PCR: bone marrow, heart, lymph nodes, liver, lungs, skin and spleen. Mainly serological test were employed in the maned wolf (*Chrysocyon brachyurus*) with prevalence values from 10% to 75% depending on the study [82,84,85,88,89,90], while the parasite was found only in bone marrow and skin by PCR [84,85]. In the bush dog (*Speothos venaticus*), several techniques were employed including serology, culture of isolates, PCR, smears and histopathology. While most studies reported results from only one or two animals, only three studies analysed a higher number (4–6) and found 33.3% positives using PCR (blood) and 60–100% of the animals positive by serology [84,85,89,90,91] (Appendix A). The potential transmission to the vector was demonstrated in manned wolves and bush dogs [89], which reinforces their role as reservoirs. 

Several studies investigated free-ranging carnivores by serology using a direct agglutination test (DAT), and positive values were found in tayras (*Eira barbara*), lesser grison (*Galictis cuja*) and coatis (*Nasua nasua*) in Brazil at high serum dilutions (≥1:1280) [92]. In carnivores kept in captivity, serology was also employed to demonstrate the presence of antibodies against the parasites in ocelots (*Leopardus pardalis*), hoary foxes (*Lycalopex*–*Pseudalopex vetulus*), jaguars (*Panthera onca*), Siberian tigers (*Panthera tigris altaica*), African lions (*Panthera leo*) and cougars (*Puma concolor*) (Appendix A) [84,90,93,94]. Clinical signs of VL were more frequent in wild canids compared to wild felines and included weight loss, anaemia, lymph node enlargement, vomiting, diarrhoea and polydipsia/polyuria, which were described in some of the animals from the previously mentioned species, such crab-eating foxes [85], bush dogs [84,85], hoary foxes [84], Siberian tigers and maned wolves [84,90]. African lions were reported to test positive for the first time by PCR (kDNA) followed by RFLP, but the animal did not show clinical signs [94]. Finally, *Leishmania* infection (probably *L. infantum*) was found in the kidney of one road-killed crab-eating raccoon (*Procyon cancrivorus*) in Brazil by PCR followed by sequencing [86].

The DNA of *L. infantum* was found in at least 17 species of bats in nine studies, including one hematophagous species and several frugivorous, omnivorous or insectivorous ones [14,32,44,45,66,95,96,97,98]. The feeding habits of the animals were relevant, since the oral route was suggested for transmission in animals feeding on insects, including the vector of leishmaniasis [14]. They also shared the same ecological niche in bat caves and probably other locations. Values of infection varied widely, from less than 1% to 100% of the analysed bats, being infected mainly in the blood but also in the liver, skin, spleen and even in oral swab samples [32]. PCR followed by RFLP or sequencing was employed in the studies and, when sensitive primers were employed, a prevalence higher than 30% was usually obtained [95].

A small number of species of the orders Cingulata and Pilosa were found to be infected with *L. infantum* in Brazil. The lesser anteater (*Tamandua tetradactyla*) was reported to test positive by PCR (kDNA) in blood and bone marrow in 2013 [99] and again in 2014, together with giant anteaters (*Myrmecophaga tridactyla*) and one seven-banded armadillo (*Dasypus septemcinctus*) found dead on the roads in Brazil, employing PCR from several tissues [86]. 

Marsupials were studied in several surveys by PCR or serology (Appendix A). The white-eared opossum (*Didelphis albiventris*) was analysed by six groups in Brazil, who found the parasite in blood, bone marrow, lungs, kidney, skin and spleen by culture or PCR (kDNA, ITS1 or SSU) and sequencing or RFLP [38,39,50,54,86,100,101], with percentages of infection between 6.3% and 22.2%. The big-eared opossum (*Didelphis aurita*) was positive at a low percentage in Brazil by PCR, spleen imprints and serology (rK39), and one of the animals displayed spleen enlargement, but no other clinical signs were recorded from the rest [102]. In Brazil as well as in Colombia and Venezuela, the common opossum (*Didelphis marsupialis*) was widely analysed [102,103,104,105,106,107]. Two studies in Colombia demonstrated the transmission of isolates from common opossums to hamster, highlighting their role as reservoirs of *L. infantum* [105,106]. The parasite was found in several tissues employing PCR (kDNA, SSU and ITS1) followed by hybridisation or RFLP [38,103,107]. Two studies in Brazil employed serology and PCR simultaneously. In the first one, the authors found 9–21.6% of the animals positive using serology, and only 5% positive by PCR–RFLP [38], while the other study analysed 112 individuals of two species (i.e., white-eared and big-eared opossums), and found high percentages of positivity (71–91.6%) with both techniques (see Appendix A for details). 

Lagomorphs were scarcely reported as exposed to *L. infantum* in the Americas, with one European hare (*Lepus europaeus*) found positive in Brazil by DAT and with a low antibody titre (1:320) [92].

Infection with *L. infantum* in primates were studied in five surveys, and eleven species were reported with DNA of the parasite. Several species of captive primates showed high prevalence values when employing PCR (kDNA) in an endemic area of Brazil including brown howler monkeys (*Alouatta guariba*), black-headed night monkeys (*Aotus nigriceps*), black-fronted titi (*Callicebus nigrifons*), golden-bellied capuchin (*Cebus xanthosternos*), golden-headed lion tamarin (*Leontopithecus chrysomelas*), bald-faced saki (*Pithecia irrorata*) and emperor tamarin (*Saguinus imperator*). Among them, one black-fronted titi was found dead with clinical signs compatible with leishmaniasis, but the rest did not show clinical signs [108]. On the other hand, free-ranging howler monkeys (*Alouatta caraya*) sampled at the marginal area of an endemic region from Argentina displayed low values of prevalence (6.3%) [64]. Two other studies found indirect evidence of infection with the parasite using DAT in one white-tufted-ear marmoset (*Callithrix jacchus*) [109] and 26.9% of the black-tufted marmosets (*Callithrix penicillata*) [109]; the last study also employed PCR of the skin. Positive serology was detected in twenty-two percent of red howler monkeys (*Alouatta seniculus*) in French Guiana, and data were further confirmed by PCR (110).

Rodents occupied most of the attention of researchers investigating *L. infantum* in the Americas, and twelve studies fulfilled the inclusion criteria of this systematic review [21,36,39,50,53,58,86,92,101,106,107,110]. Most of the studies employed different PCR approaches in several tissues, although in one study the authors detected antibodies. Brazilian guinea pigs were reported to be infected in Brazil by PCR in heart tissue [86]. The infection was also found in two species of porcupines from Brazil: the prehensile tailed porcupine (*Coendu–Sphiggurus villosus*) by serology (DAT) [92], which is indirect evidence of the parasitism, and the Paraguayan hairy dwarf porcupine (*Coendou–Sphiggurus spinosus*) by PCR and sequencing from several tissues Appendix A) [86]. Agouties were also reported to test positive for *L. infantum* in the spleen (16.7%), skin and blood by PCR [36,110]. The giant rodent capybara (*Hydrochoerus hydrochaeris*) was positive in the lungs by PCR and sequencing [86]. Several species from wild mice, rats and cricetidae of the genera *Cerradomys*, *Clyomis*, *Holochilus*, *Hylaeamys, Nectomys*, *Oryzomys*, *Proechymis*, *Rhipidomys* and *Trichomys* were reported to be infected with *L. infantum* in several surveys, and in previous studies the potential role as reservoir of some of them was indicated [24]. The authors employed distinct approaches of PCR followed by RFLP, hybridisation or sequencing [21,36,39,50,58,106,110]. 

Synanthropic rodents, such as the house mouse (*Mus musculus*), the black rat (*Rattus rattus*) and the brown rat (*Rattus norvegiccus*) were investigated in Brazil and Venezuela. Researchers found 20% of house mice to be infected [50], while the prevalence in black rats varied widely, with values from 0.1% to 100% using several approaches of PCR followed by hybridisation, RFLP or sequencing [21,39,50,53,58,107]. Almost 17% of brown rats were positive by nPCR and sequencing [50]. These synanthropic species of rodents could act as relevant reservoirs of leishmaniasis, since they were infected at high percentages and share habitats with humans.

Additional species were analysed in other surveys or in some of the previous studies in which the authors could not characterise the parasite at the species level (Appendix A) [44,91,111,112,113,114,115,116,117,118]. Indirect evidence of *Leishmania* spp. was found in the USA using rapid tests (rk30 antigen) in several wild carnivores such as coyotes (*Canis latrans*, 1.6%), American red foxes (*Vulpes fulvus*, 9.1%) and grey foxes (*Urocyon cinereoargenteus*, 2%) [111,112]. On the other hand, DNA of *Leishmania* spp. was detected in several species, including carnivores, such as South American grey foxes (*Lycalopex–Pseudalopex griseus*) [113] and Neotropical otters (*Lontra longicaudis*), and some species of primates, such as black-headed night monkeys (*Aotus nigriceps*), black-bearded sakis (*Chiropotes satanas*) and grey-woolly monkeys (*Lagothrix cana*) in Brazil [114]. Among rodents, *Leishmania* spp. DNA was reported in the blood of a red-tailed squirrel (*Sciurus granatensis*) in Venezuela [103]. All this information is summarized in Table 5.

The following species have been reported to be infected with *L. infantum* in the Americas in previous reviews: the rodent *Proechymis spinosus*, the Brazilian porcupine (*Coendu prehensilis*) and the fennec fox (*Vulpes zerda*). More information can be retrieved from the abovementioned reviews [23,24,25,27].

#### 3.5.2. *L. infantum* in Wild Animals from Europe, Asia and Africa

Carnivores, bats, wallabies, hedgehogs, lagomorphs and rodents tested positive for *L. infantum* via antibody detection or PCR in several countries in South Europe (Croatia, France, Greece, Italy, Portugal, Spain, Romania), North Africa (Morocco, Tunisia) and Asia (Georgia, Iran, Israel, Saudi Arabia), Spain being the country with the highest number of studies of wild animals (Appendix A). 

*L. infantum* infection was reported in a large list of wild carnivores including 18 species. The golden jackal (*Canis aureus*) was positive in four studies at low percentages. The lowest values were found in Georgia using a rapid test (2.6%) [120] and in Romania employing PCR and sequencing of the ITS1 region from bone marrow samples (3%) [121]. In Iran, 11.6% of the jackals were found positive by serological test (DAT), and subsequent smears and culture from skin lesions, lymph nodes, spleen and liver were furthered characterised by PCR and sequencing [122]. Only 7.8% of the animals were found positive by PCR using blood samples [123]. The target or the technique used in each study could influence the results obtained by the different authors.

The wolf was the focus of research of many zoonoses including *L. infantum* infections. In this review, nine studies reported the animal to be positive for the parasite, although three of them included a low number of animals (three or less). In Croatia, only one wolf was reported positive by PCR and sequencing of the cysteine protease B in lymph nodes [124]. The remaining authors employed PCR of the kDNA region to detect the infection and, in some cases, RFLP and sequencing were later applied. Prevalence values from 33% to 50% were found in Spain when using spleen, skin or lymph nodes as samples [125,126,127,128,129] including one study conducted in a non-endemic region [127]. The percentage of infected animals was lower when blood (9%) [130] or hair (4.1%) [131] were used to detect the parasite. In Italy, 25% of the animals were infected when samples from spleen were analysed [132]. Skin lesions were reported only in one study [127]. 

A smaller number of individuals from wild cats (*Felis silvestris*) [127,128,133] and genets (*Genetta genetta*) [125,127,128,129,133,134] tested positive to *L. infantum* in samples from skin, liver or spleen employing PCR of the kDNA. The percentage of infection in both species reached 100% of the sampled animals, although wild cats showed 25% as the minimum value [128,133], while genets displayed a 10% prevalence in blood or spleen samples [134]. In two studies, additional PCR and sequencing of the ITS2 region were carried out [129,133].

Several studies were conducted employing similar approaches to detect *L. infantum* in various species of carnivores. The parasite was found in the spleen or blood of 28.6% of Egyptian mongoose (*Herpestes ichneumon*) from Spain [125] and 4.7% (only spleen) from Portugal [135]. Seventy percent of otters’ (*Lutra lutra*) spleens [136] and 25% of Iberian lynxes’ (*Lynx pardinus*) samples (spleen and blood) [135] were found to be infected in two studies. Six surveys reported the presence of *L. infantum* in hair, liver, spleen or lymph nodes of 29–100% of sampled beech martens (*Martes foina*) in Spain [127,128,129,131,133,137]. Values between 30% and 62% were found in pine martens (*Martes martes*) using the same techniques [127,133,134]. Badgers (*Meles meles*) were found infected in the liver or spleen in Italy (53%) [132] and Spain (26%) [133]. European minks (*Mustela lutreola*) were found to be infected with values of 50% in Spain using a similar methodology [133] but at lower percentages (e.g., 2.1%) when ELISA or PCR of the ITS1 were employed in Greece [136]. Moreover, 20% of pole cats (*Mustela putorius*), 45% of tigers (*Panthera tigris*) in a zoo, 20% of red squirrels (*Sciurus vulgaris*) and one individual of each species of American mink (*Mustela vison*) and brown bear (*Ursus arctos*) were reported to be infected with the protozoa [129,133,137,138].

The DNA of *L. infantum* was found in at least 17 species of bats in nine studies, including one hematophagous species and several frugivorous, omnivorous or insectivorous ones [14,32,44,45,66,95,96,97,98]. The feeding habits of the animals were relevant, since the oral route was suggested for transmission in animals feeding on insects, including the vector of leishmaniasis [14]. They also shared the same ecological niche in bat caves and probably other locations. Values of infection varied widely, from less than 1% to 100% of the analysed bats, being infected mainly in the blood but also in the liver, skin, spleen and even in oral swab samples [32]. PCR followed by RFLP or sequencing was employed in the studies and, when sensitive primers were employed, a prevalence higher than 30% was usually obtained [95].

A small number of species of the orders Cingulata and Pilosa were found to be infected with *L. infantum* in Brazil (Appendix A). The lesser anteater (*Tamandua tetradactyla*) was reported to test positive by PCR (kDNA) in blood and bone marrow in 2013 [99] and again in 2014, together with giant anteaters (*Myrmecophaga tridactyla*) and one seven-banded armadillo (*Dasypus septemcinctus*) found dead on the roads in Brazil, employing PCR from several tissues [86]. 

Marsupials were studied in several surveys by PCR or serology (Appendix A). The white-eared opossum (*Didelphis albiventris*) was analysed by six groups in Brazil, who found the parasite in blood, bone marrow, lungs, kidney, skin and spleen by culture or PCR (kDNA, ITS1 or SSU) and sequencing or RFLP [38,39,50,54,86,100,101], with percentages of infection between 6.3% and 22.2%. The big-eared opossum (*Didelphis aurita*) was positive at a low percentage in Brazil by PCR, spleen imprints and serology (rK39), and one of the animals displayed spleen enlargement, but no other clinical signs were recorded from the rest [102]. In Brazil as well as in Colombia and Venezuela, the common opossum (*Didelphis marsupialis*) was widely analysed [102,103,104,105,106,107]. Two studies in Colombia demonstrated the transmission of isolates from common opossums to hamster, highlighting their role as reservoirs of *L. infantum* [105,106]. The parasite was found in several tissues employing PCR (kDNA, SSU and ITS1) followed by hybridisation or RFLP [38,103,107]. Two studies in Brazil employed serology and PCR simultaneously. In the first one, the authors found 9–21.6% of the animals positive using serology, and only 5% positive by PCR–RFLP [38], while the other study analysed 112 individuals of two species (i.e., white-eared and big-eared opossums), and found high percentages of positivity (71–91.6%) with both techniques (see Appendix A for details). 

Lagomorphs were scarcely reported as exposed to *L. infantum* in the Americas, with one European hare (*Lepus europaeus*) found positive in Brazil by DAT and with a low antibody titre (1:320) [92].

Infection with *L. infantum* in primates were studied in five surveys, and eleven species were reported with DNA of the parasite. Several species of captive primates showed high prevalence values when employing PCR (kDNA) in an endemic area of Brazil including brown howler monkeys (*Alouatta guariba*), black-headed night monkeys (*Aotus nigriceps*), black-fronted titi (*Callicebus nigrifons*), golden-bellied capuchin (*Cebus xanthosternos*), golden-headed lion tamarin (*Leontopithecus chrysomelas*), bald-faced saki (*Pithecia irrorata*) and emperor tamarin (*Saguinus imperator*). Among them, one black-fronted titi was found dead with clinical signs compatible with leishmaniasis, but the rest did not show clinical signs [108]. On the other hand, free-ranging howler monkeys (*Alouatta caraya*) sampled at the marginal area of an endemic region from Argentina displayed low values of prevalence (6.3%) [64]. Two other studies found indirect evidence of infection with the parasite using DAT in one white-tufted-ear marmoset (*Callithrix jacchus*) [109] and 26.9% of the black-tufted marmosets (*Callithrix penicillata*) [109]; the last study also employed PCR of the skin. Positive serology was detected in twenty-two percent of red howler monkeys (*Alouatta seniculus*) in French Guiana, and data were further confirmed by PCR (110).

Rodents occupied most of the attention of researchers investigating *L. infantum* in the Americas, and twelve studies fulfilled the inclusion criteria of this systematic review [21,36,39,50,53,58,86,92,101,106,107,110]. Most of the studies employed different PCR approaches in several tissues, although in one study the authors detected antibodies. Brazilian guinea pigs were reported to be infected in Brazil by PCR in heart tissue [86]. The infection was also found in two species of porcupines from Brazil: the prehensile tailed porcupine (*Coendu–Sphiggurus villosus*) by serology (DAT) [92], which is indirect evidence of the parasitism, and the Paraguayan hairy dwarf porcupine (*Coendou–Sphiggurus spinosus*) by PCR and sequencing from several tissues (Appendix A) [86]. Agouties were also reported to test positive for *L. infantum* in the spleen (16.7%), skin and blood by PCR [36,110]. The giant rodent capybara (*Hydrochoerus hydrochaeris*) was positive in the lungs by PCR and sequencing [86]. Several species from wild mice, rats and cricetidae of the genera *Cerradomys*, *Clyomis*, *Holochilus*, *Hylaeamys, Nectomys*, *Oryzomys*, *Proechymis*, *Rhipidomys* and *Trichomys* were reported to be infected with *L. infantum* in several surveys, and in previous studies the potential role as reservoir of some of them was indicated [24]. The authors employed distinct approaches of PCR followed by RFLP, hybridisation or sequencing [21,36,39,50,58,106,110]. 

Synanthropic rodents, such as the house mouse (*Mus musculus*), the black rat (*Rattus rattus*) and the brown rat (*Rattus norvegiccus*) were investigated in Brazil and Venezuela. Researchers found 20% of house mice to be infected [50], while the prevalence in black rats varied widely, with values from 0.1% to 100% using several approaches of PCR followed by hybridisation, RFLP or sequencing [21,39,50,53,58,107]. Almost 17% of brown rats were positive by nPCR and sequencing [50]. These synanthropic species of rodents could act as relevant reservoirs of leishmaniasis, since they were infected at high percentages and share habitats with humans.

Additional species were analysed in other surveys or in some of the previous studies in which the authors could not characterise the parasite at the species level (Supplementary Material File S4) [44,91,111,112,113,114,115,116,117,118]. Indirect evidence of *Leishmania* spp. was found in the USA using rapid tests (rk30 antigen) in several wild carnivores such as coyotes (*Canis latrans*, 1.6%), American red foxes (*Vulpes fulvus*, 9.1%) and grey foxes (*Urocyon cinereoargenteus*, 2%) [111,112]. On the other hand, DNA of *Leishmania* spp. was detected in several species, including carnivores, such as South American grey foxes (*Lycalopex–Pseudalopex griseus*) [113] and Neotropical otters (*Lontra longicaudis*), and some species of primates, such as black-headed night monkeys (*Aotus nigriceps*), black-bearded sakis (*Chiropotes satanas*) and grey-woolly monkeys (*Lagothrix cana*) in Brazil [114]. Among rodents, *Leishmania* spp. DNA was reported in the blood of a red-tailed squirrel (*Sciurus granatensis*) in Venezuela [103].

Seventeen studies reported the infection in the red fox (*Vulpes vulpes*), with lower prevalence values found in France (9–15%) and Georgia (2.6%) and higher in southern European Mediterranean countries, such as Italy (12.3–40%), Greece (59.5%) or Spain (12–74%) (Appendix A) [122,125,127,128,129,131,132,139,140,141,142,143,144,145,146]. The high numbers of publications might be due to the epidemiological relevance of this animal because it is a widespread species, which inhabit different ecosystems where the parasite life cycle can be completed, from forest to areas close to human settlements. One study carried out in Iran reported intermediate values (28.6%) using serology, cultures, smears and PCR-sequencing from lymph nodes, skin and spleen [122]. Serology (ELISA, IFAT) was first employed in Italy [139] and then in Georgia (recombinant antigen rK39 rapid test) [120], but since 2000, PCR was the most widely used technique to detect the infection [122,125,127,128,129,131,132,140,141,142,143,145,146]. Although no clinical signs were reported in most of the studies, the majority of the animals from the study carried out in Greece (63.8%) showed at least 2–3 clinical signs compatible with canine VL including low weight, dermatitis, skin lesions, alopecia, esplenomegaly, enlargement of lymph nodes and onychogryphosis [143]. 

The infection was demonstrated also in Bennett’s wallabies in a zoo in Madrid (Spain) using serology (rk39) and PCR followed by sequencing of the ITS1 and ITS2 regions. Thirty- three percent of the animals were infected in several tissues including blood, kidneys, lymph nodes, liver, lungs, skin and spleen [147]. In addition, two North West Bornean orangutan (*Pongo pygmaeus*) with clinical signs of visceral leishmaniasis were found to be infected using PCR (ITS1) of the bone marrow and serology [148].

Over the last decade, new reservoirs of leishmaniasis in Europe have been reinforced through investigation with bats, lagomorpha and hedgehog species, enlarging the list of wild animals infected with the parasite. Only one study in Europe demonstrated the presence of *L. infantum* in spleen, hair and blood of 51.9% common urban bats (*Pipistrellus pipistrellus*) in Spain, using PCR and sequencing of the repeat region [149]. One hundred percent of Algerian hedgehogs were found infected in two studies in Tunisia [150,151]. The authors employed smears, PCR and RFLP and sequencing of several targets. Spleen, liver, kidney, heart, lymph nodes, blood and eye swabs were positive to all of the techniques employed. In Spain, the European hedgehog was found to be infected using ELISA and qPCR of the kDNA region, finding higher values of infection in the spleen than in skin samples [137]. The parasite was also detected in a hair sample from one animal [131].

The DNA of *L. infantum* was first detected in 2013 in the spleen of European hares (*Lepus europaeus*) and Iberian hares (*Lepus granatensis*) from Spain, with 43.6% of the animals infected [152], since the outbreak of human leishmaniasis in Madrid motivated research on wild reservoirs. Since then, other studies were conducted finding the parasite in spleen, hair and skin with molecular (PCR of several targets), histological (direct antibody fluorescence assay) and serological analyses (IFAT) in Spain [153,154], Italy [155,156] and Greece [136]. Similar approaches were used for the European rabbit (*Oryctolagus cuniculus*), which was investigated in six studies from Spain [128,129,153,154,157,158]. The authors found positive values ranging from 0.6% to 59%, depending on the time of the year, the sampling area or the techniques employed (i.e., smears, ELISA, IFAT, PCR). In general, serology displayed lower percentages of infection than PCR. In Italy and Greece, lower values of infected animals were obtained, but as previously pointed out, this might be due to the techniques employed (serology or PCR of ITS) [136,145]. 

Wild and synanthropic rodents were always the focus of researchers interested in *L. infantum* epidemiology, similar to what happened with other species of *Leishmania*. In Spain, the wood mouse (*Apodemus sylvaticus)* was positive with values of 20–50%, depending on the study, in several tissues [128,129,159]. The authors employed culture and smears [159], but also PCR followed by RFLP or sequencing of ITS1, ITS2, kDNA and SSU regions. Blood and spleen from shrews (*Crocidura russula*) were found to be infected at a low prevalence [160] as well as Algerian mice (*Mus spretus*), in percetanges from 4.3% to 42.9% using PCR in several tissues as well as serology [137,160]. In Iran, 39% of the sampled shor-tailed bandicoot rat (*Nesokia indica*) were positive in smears of skin and spleen, and later characterised as *L. infantum* by nested PCR of the kDNA when compared with reference isolates [161].

Synanthropic rodents were studied in eight surveys displaying high prevalence values for *Leishmania* infection. In most of them, *L. infantum* was found in the skin, liver and spleen of house mice (*Mus musculus*) and brown rats (*Rattus norvegicus*) in Portugal and Morocco [162,163], with animals displaying skin lesions in both studies. Smears were employed in Portugal while PCR and sequencing were employed in both studies, although different targets were selected (kDNA, ITS1, SSU). Prevalence values ranged from 22% to 33.3% in mice and 33% in rats. In Spain, 50% of house mice were found to test positive using PCR and sequencing of blood, bone-marrow and skin samples [159]. Brown rats were found infected using similar methologies, with 33–100% of animals positive in Spain [129,131] and 5.9% in Greece [164]. Black rats (*Rattus rattus*) were found infected in 18.2% of sampled animals in Saudi Arabia [163], 15.5% in Italy [165], 33.3% in Spain [157] and 7.5% in Morocco [161].

A summary of the techniques and organs found infected wit *L. infantum* in wild animals from Europe, Asia and Africa is summarised in Table 6.

In previous reviews, other species were reported infected with *L. infantum* or were described as clinical cases: domestic ferrets (*Mustela putorius furo*), corsac foxes (*Vulpes corsak*), raccoon dogs (*Nyctereutes procyonoides*), Mediterranean monk seals (*Monachus monachus*), Persian jirds (*Meriones persicus*), Syrian hamsters (*Mesocricetus auratus*), grey hamsters (*Cricetulus migratorius*) and porcupines (*Hystrix* sp.). Detailed information can be found in specific reviews [19,26,27].

### 3.6. Wild Animals Infected with L. major

*L. major* infections extended through Asian and African countries [3,5], and nineteen studies were carried out from 1990 in wild animals in Algeria, Tunisia, Iran, Israel, Ethiopia, Kenya, Cameroon and Morocco (Appendix A).

In the included studies, the most employed techniques to detect *L. major* were PCR of the kDNA (nine studies) and ITS (nine studies) regions, although SSU and the repeat regions were also employed, mainly followed by RFLP and/or sequencing (four and two studies, respectively). Only seven studies exclusively employed the skin to search for the parasite, but the rest of the studies employed also other anatomical sites such as liver, spleen, heart, blood, kidney, lymph nodes, eye swabs or even feces. Noteworthy were the higher prevalence values observed in smears compared with the PCR of the kDNA in some studies, probably due to the methodology employed, since the DNA was extracted from fixed smears [161,167,168,169,170,171] (for details, see Appendix A).

The infection was demonstrated in the orders Chiroptera, Eulipotyphla, Primates, and Rodentia, the last group, again, being the most widely studied. Only one species of bat (*Nycteris hispida*) was reported in Ethiopia to be infected with *L. major* in the spleen by qPCR and sequencing of the kDNA and ITS regions [172]. The DNA of the parasite was found in several organs and tissues in three species of hedgehogs in Algeria, Iran and Tunisia including spleen, skin, heart, kidney, liver, blood and eye swab. A hundred percent prevalence was reported in two studies carried out on the Algerian hedgehog (*Atelerix algirus*) in Tunisia [150,151], while 36.8% was reported in Algeria by serology and PCR–RFLP of the kDNA region employing the spleen and skin [173] (Appendix A). Two studies, including the long-eared hedgehog (*Hemiechinus auritus*), reported prevalence rates ranging from 33%, using nPCR of the ITS and smears of the skin [173], to 53.3%, employing nPCR of the kDNA from the skin and smears from skin, liver and spleen [174]. The desert hedgehog (*Paraechinus aethiopicus*) was found to be infected with *L. major* in two studies. The first study employed qPCR of the kDNA region from spleen and skin as well as serology [175], and the other reported the infection in the kidney, blood, liver, eye swab and lymph node of one animal by qPCR of the kDNA, SSU and repeat regions [151]. The high values of infection found in these animals suggest their role as reservoirs, and these animals should be monitored in endemic areas.

One study investigated the immune response to *L. major* of three species of primates in Kenya (*Cercopithecus mitis*, *Chlorocebus aethiops* and *Papio cynocephalus anubis*) including humoral (ELISA and Western blot) and cellular responses (lymphoproliferative assay) [176]. The authors include 57–213 individuals per technique and found that 60–77% of the animals were previously exposed to the parasite. Surprinsingly, one study found parasites (amastigote and promastigotes forms) and DNA of *L. major* in the feces of gorillas (*Gorilla gorilla*) in Cameroon, and the authors pointed to the ingestion of phlebotomines by the animals [15]. However, no other method to measure exposure (serology) or presence of the parasite in organs was employed.

Eleven species of rodents were reported as infected with the parasite in nine studies from Iran, and another one from Israel (Appendix A), following the same tendency of other zoonotic species of *Leishmania* included in the systematic review. The higher prevalence was found in *Meriones libycus* by PCR of kDNA in the skin [177], which is also the species most studied regarding *Leishmania* infections in Iran [167,168,169,170,171,173,177,178]. Other species of *Meriones* (*M. hurrianae*, *M. persicus* and *M tristrami*) and *Microtus* (*M. guentheri* and *M. socialis*) were reported to be infected with parasites with values of 5.7–58.3% [161,168,171,179]. *Mus musculus* was found to be infected with low percentages of infection (2.3–33%) in three studies carried out in Iran and Israel. However, PCR of the ITS region from skin samples, and smears from skin, liver and spleen, were used instead of PCR of kDNA, [161,179]. *Nesokia indica* was found to be infected in three studies from Iran, ranging from 8% in skin by smears and PCR of the ITS region [168] to values higher than 61% employing smears and PCR of skin, liver and spleen tissues [161,167]. The same techniques (PCR of the ITS and kDNA regions) were employed in three studies in Iran to detect *L. major* infections in *Tatera indica* from skin, liver or spleen, with values of prevalence ranging from 3,7 to 50% [168,171,178]. The great gerbil (*Rhombomys opimus*) was reported infected with the parasite in Iran [168,178] using smears, PCRs and inoculation of hamsters (see Appendix A for details). The high prevalence found in many of these rodent species points to their reservoir role. 

Most of the studies did not find clinical signs in infected animals, or the authors did not look for them; however, skin lesions were recorded in *Meriones libycus* from Iran [177]. 

All the information concerning *L. major* infection in wild animals is summarised in Table 7.

In previous reviews, other species were reported infected with *L. major* in the past including primates (*Cercopithecus aethiops*) and rodents (*Xerus rutilus*, *Gerbillus pyramidum*, *Tatera gambiana*, *Tatera robusta*, *Taterillus emini*, *Meriones crassus*, *Meryones meridianus*, *Meryones shawi*, *Psammomis obesus*, *Praomys erythroleucus* and *Mastomys natalensis*) [23].

### 3.7. Wild Animals Infected with Leishmania tropica

Four studies conducted in Ethiopia [175,180], Kenya [181] and Egypt [182] demonstrated the presence of *L. tropica* DNA in the spleen of rodents and bats. One hundred and sixty-three bats (*Cardioderma cor*) were analyzed in Ethiopia by qPCR of the kDNA and ITS1 regions and 4.9% were found to be infected [175]. Using the same techniques, the authors found prevalences from 9.9% to 20% in the rodents *Acomys* sp. *Arvicanthis niloticus* and *Gerbillus nanus* [180] (Appendix A). In Egypt, *L. tropica* was found in 14.3% of the analyzed Anderson’s gerbils (*Gerbillus andersoni*) with clinical signs by smears and PCR of the ITS1 region [182]. Finally, employing nested PCR and sequencing of the SSU and the ITS1 regions, 22% of the sampled house mice (skin) were found infected with *L. tropica* in Morocco [163].

The species *Procavia capensis* and *Arvicanthis niloticus* were reported in previous reviews infected with *L. tropica* [23].

### 3.8. Wild Animals Infected with Leishmania donovani

In Africa, rodents were infected with *L. donovani* in three studies. One employed serology and found 5.5% of African grass rats (*Arvicanthis niloticus*) to be positive by ELISA in Sudan [183], while another study found 18.2% of black rats (*Rattus rattus*) with clinical signs carrying the parasite in Saudi Arabia [164], employing culture, smear and inoculation of hamster. The other study found DNA of the *L. donovani* complex in the spleen in 15.3% of *Mastomys erythroleucus*, 7.7% of *Gerbilliscus nigricaudus* and 17.4% of *Arvicanthis niloticus* from Ethiopia using PCR of the kDNA and ITS regions [180].

Finally, one study obtained a prevalence of 23.5% of the *L. donovani* complex in European hares from Greece by nested PCR of the ITS region, employing the spleen of 166 animals [184] (Appendix A).

Details of infection with *L. donovani* of *Felis serval* can be found in a previous review [23].

Some of the previously mentioned studies obtained positive results for *Leishmania* spp., but they could not further determine the species (see Appendix A). The spleen of other four species of bats (*Glauconycteris variegate*, *Miniopterus arenarius*, *Neoromicia somalica* and *Scotophilus colias*) were found to be infected with *Leishmania* spp. in Ethiopia by PCR [175]. In a similar approach, 40% of the analyzed rodents of *Aethomys* spp. were also positive [180]. Smears of liver and spleen and indirect haemagglutination test were employed to detect the parasite in 40% of the sand cat (*Felis margarita*) in Saudi Arabia [185]. The same techniques were employed to detect *Leishmania* spp. infections in *Gerbillus pyramidum* and *Rattus norvegicus* in Egypt [186].

In previous reviews, other species of mammals, such as *Crycetomys gambianus, Heterohyrax brucei* and *Dendrohyrax arboreus*, were reported to be infected with *L. aethiopica* [23]. 

## 4. Conclusions

Knowledge of the role of wild animals as suitable hosts or reservoirs of *Leishmania* zoonotic species is essential in order to apply control measues or monitoring programmes. In this review, a systematic search of wild animals infected with zoonotic species of *Leishmania* was conducted, starting from 1990 and following PRISMA methodology. One hundred and eighty-nine species of wild animlas from ten orders (i.e., Carnivora, Chiroptera, Cingulata, Didelphimorphia, Diprotodontia, Lagomorpha, Eulipotyphla, Pilosa, Primates and Rodentia) were included in the review. Rodents and carnivores were the orders more widely explored, being the most probable main reservoirs, and also the ones presenting more clinical signs. *L. infantum* was the most widely distributed species, both geographically and in the range of species, followed by *L*. (*Viannia*) *braziliensis*, but this fact could be due to the more exhaustive investigation on these species.

More studies on the role of infected wild animals are necessary in order to implement specific measures when an outbreak of the disease appears.

## Figures and Tables

**Figure 1 microorganisms-09-01101-f001:**
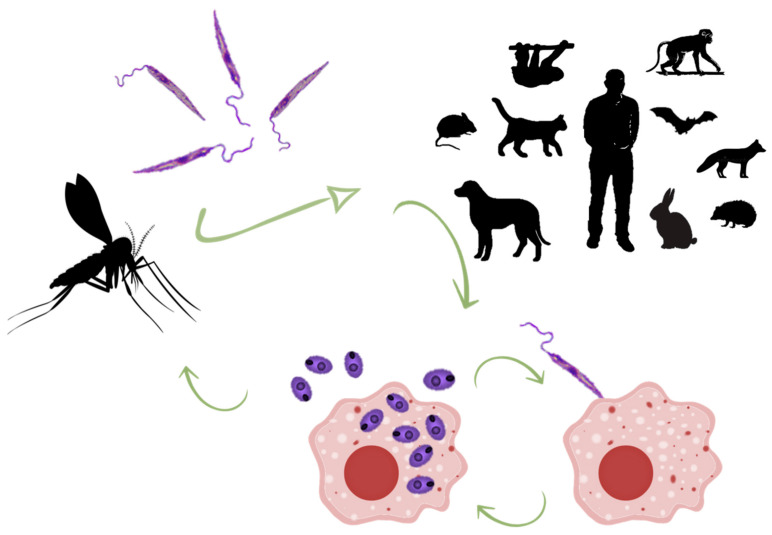
Life cycle of *Leishmania*: some of the wild animals found infected with the parasite are included.

**Figure 2 microorganisms-09-01101-f002:**
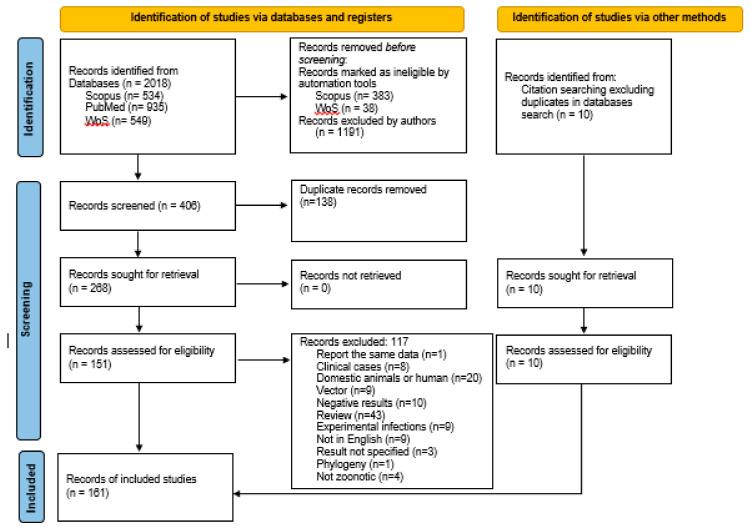
PRISMA 2020 flow diagram for the present systematic review.

**Table 1 microorganisms-09-01101-t001:** Zoonotic species of the genus *Leishmania* and their main characteristics (adapted from Ahoundi et al., 2016 and 2017 [4,5]).

Section	Subgenus	Species	Clinical Disease in humans	Geographic Area
Euleishmania	*Leishmania*	***L. aethiopica***	CL, DCL	Ethiopia, Kenya
		***L. amazonensis***	CL, DCL, MCL	Bolivia, Brazil and Venezuela
		***L. donovani***	VL, PKDL	Central Africa, South Asia, Middle East, India and China
		***L. infantum***	VL, CL	North Africa, South Europe, Middle East, Central Asia and North, Central and South America
		***L. major***	CL	Central and North Africa, Middle East and Central Asia
		***L. mexicana***	CL, DCL	USA, Mexico, Ecuador, Peru and Venezuela
		***L. tropica***	CL, VL	Central and North Africa, Middle East, Central Asia and India
		***L. venezuelensis***	CL	Northern South America
	*Viannia*	***L. braziliensis***	CL, MCL	Brazil, Bolivia, Peru, Guatemala and Venezuela
		***L. guyanensis***	CL, MCL	Bolivia, Brazil, French Guyana and Suriname
		***L. lainsoni***	CL	Brazil, Bolivia and Peru
		***L. lindenbergi***	CL	Brazil
		***L. naiffi***	CL	Brazil, French Guyana
		***L. panamensis***	CL, MCL	Brazil, Panama, Venezuela and Colombia
		***L. peruviana***	CL, MCL	Peru, Bolivia
		***L. shawi***	CL	Brazil
Paraleishmania		***L. colombiensis***	CL, VL	Colombia

Zoonotic species included in the systematic review are in bold. Clinical forms in humans: CL: cutaneous leishmaniasis; DCL: Diffuse cutaneous leishmaniasis; VL: visceral leishmaniasis; MCL: muco-cutaneous leishmaniasis; PKDL: Post-kala-azar dermal leishmaniasis.

**Table 2 microorganisms-09-01101-t002:** Wild animals reported infected with zoonotic *Leishmania* (*Viannia*) species. Organs or tissues where the parasite was detected are indicated, as well as the techniques employed for detection. *L* (*Viannia*) species are as follows: Lb: *L. braziliensis*, Lg: *L. guyanensis*, LVsp: *Leishmania* (*Viannia*) spp., Lsp: *Leishmania* sp., Lpa: *L. panamensis*, Lpe: *L. peruviana*, Ln: *L. naiffi*, Ls: *L. shawi* and Ll: *L. lainsoni*.

Host	Prevalence	Organs/Tissue Analysed	Methods for Detection	*Leishmania* (*Viannia*) species	Country	References
**Order Carnivora**						
*Cerdocyon thous* (crab-eating fox)	20–100%	blood, serum	PCR (kDNA), DAT	LVsp	Brazil	[34,37]
*Conepatus chinga rex* (Molina’s hog-nose skunk)	50%	Skin + liver + spleen	Inoculation to hamster, isoenzyme analysis, hybridisation, PCR (kDNA)	Lb	Bolivia	[41]
*Lycalopex (Pseudalopex) vetulus* (hoary fox)	100%	serum	DAT	LVsp	Brazil	[37]
*Nasua nasua* (ring-tailed coati)	50%	serum	DAT	LVsp	Brazil	[37]
*Procyon cancrivorous* (crab-eating raccoon)	50%	serum	DAT	LVsp	Brazil	[37]
**Order Cingulata**						
*Dasypus novemcinctus* (armadillo)	15.6%	blood, LN, liver, skin, spleen	Culture, zymodeme analysis	Lb	Brazil	[46]
*Dasypus* sp.	100%	blood	PCR (kDNA)	LVsp	Brazil	[34]
**Order Chiroptera**						
*Artibeus planirostris* (frugivorous)	4.3%	skin	PCR (kDNA), PCR (HSP70) + RFLP, PCR (G6DP) + sequencing	Lb	Brazil	[43]
*Cynomops planirostris* (insectivorous)	11.1%	liver, skin	PCR (kDNA), nPCR (SSU) + sequencing	Lb	Brazil	[44]
*Desmodus rotundus* (hematophagous)	3.2%	blood	PCR (kDNA), PCR (Cyt B) + sequencing	Lb	Brazil	[43]
*Eumops perotis* (insectivorous)	5.6%	blood	PCR (kDNA), PCR (Cyt B) + sequencing	Lb	Brazil	[45]
*Glossophaga soricina* (insectivorous)	0.9–40%	blood, liver, spleen	PCR (ITS1) + RFLP, PCR (kDNA, PCR (Cyt B) + sequencing	Lb	Brazil	[42,44,45]
*Lasiurus cinereus* (insectivorous)	20%	liver, skin	PCR (kDNA), nPCR (SSU) + sequencing	Lb, LVsp	Brazil	[44]
*Molossus molossus* (insectivorous)	44–25%	blood	PCR (ITS1) + RFLP, PCR (kDNA), PCR (Cyt b) + sequencing	Lb	Brazil	[42,45]
*Platyrrhinus lineatus* (frugivorous)	13.3%	skin	PCR (kDNA), PCR (HSP70) + RFLP, PCR (G6DP) + sequencing	Lb	Brazil	[43]
Several species: *Artibeus lituratus, Carollia perspicillata, Diphylla ecaudata and Glossophaga soricina*)	19.8%	oral swab	PCR (SSU) + sequencing	LVsp	Brazil	[32]
**Order Didelphimorphia**						
*Didelphis albiventris* (white-eared opossum)	1.6–50%	blood, BM, liver, serum, skin (tail/ear), spleen	culture, imprints, isoenzymes, PCR (kDNA), qPCR (kDNA) PCR (ITS1), PCR (HSP70), PCR (HSP70) + RFLP, PCR (ITS) + RFLP, nPCR (SSU) + sequencing, IFAT, DAT	Lb, LVsp, Lg, Lpe	Peru, Brazil	[21,30,37,38,39,47,50,51,52,53,54]
*Didelphis marsupialis* (common opossum)	20–33.3%	ear	PCR (kDNA), hybridisation, xenodiagnoses vector	Lb, LVsp	Colombia	[29,49]
*Didelphis* sp.	90%	blood	PCR (kDNA), culture	LVsp	Brazil	[34]
*Marmosa* sp.	16.7–25%	skin, spleen	PCR (kDNA), smears, culture	LVsp	Brazil	[30,39]
*Gracilinanus agilis* (agile gracile opossum)	1.4–75%	blood, BM, liver, skin (tail/ear), spleen	PCR (kDNA), PCR (HSP70), PCR (HSP70) + RFLP, PCR (ITS) + RFLP	Lb, LVsp, Lg	Brazil	[21,55]
*Marmosops incanus* (grey slender opossum)	50%	ear skin	PCR (HSP70) + RFLP		Brazil	[21]
*Micoureus demerarae* (woolly mouse opossum)	66.7%	ear	PCR (kDNA), hybridisation, xenodiagnoses vector	Lb	Colombia	[29]
*Monodelphis domestica*	25%	skin, spleen	PCR (kDNA)	LVsp	Brazil	[39]
*Micoureus pagaruayanus* (woolly-mouse opossum)	4.2–11.6%	skin	PCR (kDNA), qPCR (kDNA), nPCR (SSU), nPCR (G6DP)	Lb, LVsp	Brazil	[48]
*Micoureus* sp.	100%	blood	PCR (kDNA)	LVsp or Lsp.	Brazil	[34]
**Order Lagomorpha**						
*Sylvilagus brasiliensis* (tapeti)	100% (*n* = 1)	ear	PCR (kDNA), hybridisation, xenodiagnoses vector	Lb	Colombia	[29]
**Order Pilosa**						
*Choloepus hoffmani* (two-toed sloth)	75%	blood	Culture, PCR (kDNA), PCR (HSP70)	Lpa	Panama	[56]
**Order Primates**						
*Alouatta caraya* (black howler)	8.3%	ear tissue	PCR (ITS) + RFLP + sequencing	Lb, LVsp	Argentina	[64]
*Aotus azarai* (Azara’s night monkey)	44.4%	blood, spleen	PCR (miniexon) + RFLP + sequencing	Lb	Argentina	[65]
*Callithrix* sp.	100%	blood	PCR (kDNA)	LVsp	Brazil	[34]
*Cebus apella* (tufted capuchin)	100%	serum	DAT	LVsp	Brazil	[37]
**Order Rodentia**						
*Agouti paca* (paca)	100%	skin	culture, isoenzymes, inoculation hamster	Ll	Brazil	[57]
*Akodon arviculoides*	4%	spleen	smears, PCR (kDNA)	LVsp	Brazil	[30]
*Akodon cursor*	9.7%	liver, skin (tail), spleen	culture (liver and skin), PCR (kDNA)	Lb	Brazil	[31]
*Akodon* sp.	2.6%	blood, skin	culture + isoenzymes, PCR	LVsp	Peru	[47]
*Cerradomys* (sin. *Oryzomys*) *subflavus*	7.8–50%	BM, liver, skin (tail/ear), spleen	culture (skin), PCR (kDNA), PCR (kDNA) + hybridisation, nPCR (SSU) + sequencing, PCR (HSP70) + RFLP	Lb, LVsp	Brazil	[31,39,50,53,57]
*Calomys expulsus*	3.3%	liver	PCR (kDNA)	LVsp	Brazil	[55]
*Dasyprocta azarae* (Agouti)	75%	serum	DAT		Brazil	[37]
*Holochilus scieurus*	7.1–15%	skin, spleen	imprints, PCR (kDNA)	LVsp	Brazil	[30,39]
*Melanomys caliginosus*	21.4%	ear	PCR (kDNA) + hybridisation	Lb	Colombia	[29]
*Microryzomys minutus*	50%	ear	PCR (kDNA) + hybridisation	Lb	Colombia	[29]
*Mus musculus*	55–100%	blood, BM, liver, skin (tail/ear), spleen	PCR (kDNA) + RFLP, nPCR (SSU) + sequencing	Lb, LVsp	Brazil	[34,50,59]
*Necromys (sin. Bolomys) lasiurus*	4.9–100%	BM, liver, skin (tail/ear), spleen	culture, imprints, PCR (kDNA), PCR (kDNA) + RFLP, serodeme, isoenzyme, PCR (ITS) + RFLP, PCR (D7 24Sα rRNA = trypanosomatids) & PCR (ITS) + sequencing, nPCR (SSU) + sequencing	Lb	Brazil	[30,35,50,59,60]
*Nectomys squamipes*	7.2–28.1%	skin, spleen	culture, smears, serodeme, isoenzyme, PCR (ITS) + RFLP, PCR (kDNA), serology rK39 Ag, inoculation to hamster, zymodeme	Lb, LVsp	Brazil	[30,39,60]
*Oecomys trinitatus*	100% (*n* = 1)	ear skin	PCR (kDNA)	LV sp	Colombia	[49]
*Oligoryzomis nigripes*	26.8%	liver	PCR (kDNA)	LV sp	Brazil	[55]
*Oxymyicterus dasytrichus*	33.3%	liver	culture, PCR (kDNA), PCR (HSP70) + RFLP	Lb	Brazil	[31]
*Phyllotis andinum*	1.2%	blood, skin	culture + isoenzymes, PCR	LVsp, Lpe	Peru	[47]
*Proechymis* sp.	100% (*n* = 1)	liver, skin	PCR (kDNA)	LVsp	Brazil	[61]
*Rhipidomys macrurus*	29.6%	ear skin	PCR (D7 24Sα rRNA trypanosomatids) and PCR ITS + sequencing	Lb	Brazil	[35]
*Rattus norvegicus* (brown rat)	26.9–66.6%	blood, BM, liver, skin (tail/ear), spleen,	nPCR (SSU) + sequencing	Lb	Brazil	[50,62]
*Rattus rattus* (black rat)	2.5–50%	blood, BM, liver, skin (tail/ear), spleen	culture, hybridisation, smears, serodeme, isoenzyme, PCR (kDNA), PCR (kDNA) + RFLP, PCR (ITS) + RFLP, nPCR (SSU)+ sequencing, PCR (HSP70) + RFLP, serology rK39 Ag	Lb, LVsp	Brazil, Colombia, Venezuela	[29,30,33,39,50,53,57,60,63]
*Sigmodon hispidus* (hispid cotton rat)	0.3%-100% (*n* = 1)	blood, ear skin	culture, PCR (kDNA), PCR (kDNA) + RFLP or hybridisation	Lb, LVsp	Venezuela Colombia	[33,49]
*Trichomys apereoides*	6.3–15.6%	liver, skin (tail/ear)	PCR (HSP70) + RFLP	Lb, Lg	Brazil	[21]
*Trichomys fosteri*	2.5%	spleen	PCR (kDNA), PCR (HSP70)	Ln	Brazil	[36]
*Trichomys inermis*	3%	spleen	PCR (kDNA), PCR HSP70	Ls	Brazil	[36]
*Trichomys laurentis*	2–3.9%	spleen	PCR (kDNA), PCR (HSP70)	Lb, Ls, Ln, Lg	Brazil	[36]
*Trichomys* sp.	100% (*n* = 1)	blood	PCR (kDNA)	LVsp	Brazil	[34]
*Zygodontomys bruneus*	100% (*n* = 1)	ear skin	PCR (kDNA)	LVsp	Colombia	[49]

BM: bone marrow; Cytb: cytochrome B; DAT: direct agglutination test; FML: fucose-mannose ligand; G6DP: glucose e phosphate dehydrogenase; HSP70: heat shock protein 70 kDa; IHC: immunohistochemistry; IC: immunochromatography; ELISA: enzyme immune assay; IFAT: immunofluorescence assay; ITS: internal transcriber spacer; kDNA: kinetoplast DNA; LN: lymph node; nPCR: nested PCR; qPCR: quantitative PCR; RFLP: restriction fragment length polymorphism; SSU: small subunit of ribosomal RNA.

**Table 3 microorganisms-09-01101-t003:** Wildlife that reported positive for *Leishmania amazonensis*. Organs or tissues where the parasite was detected are indicated, as well as the techniques employed for detection.

Host	Prevalence	Organs/Tissue Analysed	Methods for Detection	Country	Reference
**Order Carnivora**					
*Conepatus chinga rex* (Molina’s hog-nose skunk)	50%	liver, skin and spleen	Inoculation to hamster, Isoenzyme typing, PCR (kDNA) PCR (trypanosomatids) + hybridisation	Bolivia	[41]
**Order Chiroptera**					
*Artibeus lituratus* (nectarivorous)	1.6%	liver, skin and spleen	nPCR (SSU), qPCR (kDNA), PCR (ITS1) + RFLP	Brazil	[14,66]
*Artibeus planirostris* (nectarivorous)	n.s.	skin	qPCR (kDNA), PCR (ITS1) + RFLP	Brazil	[14]
*Desmodus rotundus* (haematofagous)	n.s.	liver, spleen	qPCR (kDNA), PCR (ITS1) + RFLP	Brazil	[14]
*Eumops glaucinus* (insectivorous)	8.3%	liver, skin and spleen	nPCR (SSU), qPCR (kDNA), PCR (ITS1) + RFLP	Brazil	[14,66]
*Eumops auripendulus* (insectivorous)	25%	liver, spleen	nPCR (SSU)	Brazil	[66]
*Eumops perotis* (insectivorous)	5.6%	blood	PCR (kDNA), PCR (Cyt b) + sequencing	Brazil	[45]
*Glossophaga soricina* (insectivorous)	2.8–4.2%	blood, liver and spleen	nPCR (SSU), PCR (kDNA), PCR (Cyt B) + sequencing	Brazil	[45,66]
*Molossus molossus* (insectivorous)	1–1.6%	blood, liver and spleen	nPCR (SSU), PCR (kDNA), PCR (Cyt B) + sequencing	Brazil	[45,66]
*Molossus rufus* (insectivorous)	1%	liver, skin, spleen	nPCR (SSU), qPCR (kDNA), PCR (ITS1) + RFLP	Brazil	[14,66]
*Myotis nigricans* (insectivorous)	2.9%	liver, spleen	nPCR (SSU), qPCR (kDNA), PCR (ITS1) + RFLP	Brazil	[14,66]
*Nyctinomops laticaudatus* (insectivorous)	10%	liver, spleen	nPCR (SSU)	Brazil	[66]
*Phyllostomus hastatus* (omnivorous)	2.9%	blood	PCR (kDNA), PCR (Cyt b) + sequencing	Brazil	[45]
*Platyrrhinus lineatus* (omnivorous)	18.2%	blood, spleen	qPCR (kDNA), PCR (ITS1) + RFLP, PCR (kDNA), PCR (Cyt b) + sequencing	Brazil	[14,45]
*Sturnira lilium* (nectarivorous)	25%	liver, spleen	nPCR (SSU)	Brazil	[66]
**Order Didelphimorphia**					
*Marmosa (Micoureus) paraguayanus* (woolly-mouse opossum)	1.1%	skin	PCR (kDNA), qPCR (kDNA), nPCR (SSU), PCR (G6PD), sequencing	Brazil	[48]
**Order Primata**					
*Alouatta caraya* (black howler monkey)	2.8%	ear tissue	PCR (ITS) + RFLP + sequencing	Argentina	[64]
*Ateles paniscus* (spider monkey)	100%	blood	PCR (kDNA), PCR (ITS) + RFLP	Brazil	[67]
**Order Rodentia**					
*Akodon spp.*	7.1%	blood	PCR (kDNA) + hybridisation	Bolivia	[41]
*Necromys* (sin. *Bolomys*) *lasiurus*	20%	ear skin	PCR-D7 24Sα rRNA (trypanosomatids) and PCR (ITS) + sequencing	Brazil	[35]
*Oligoryzomys* spp. (rice rat)	25%	blood	PCR (kDNA) + hybridisation	Bolivia	[41]
*Hylaeamys (Oryzomys) acritus*	33.3%	tail skin	PCR (kDNA) + sequencing	Bolivia	[68]
*Oryzomys nitidus*	13.3%	tail skin	PCR (kDNA) + sequencing	Bolivia	[68]

BM: bone marrow; Cyt B: cytochrome B; G6DP: glucose 6 phosphate dehydrogenase; ITS: Internal transcriber spacer; kDNA: kinetoplast DNA; nPCR: nested PCR; n.s.: not specified; qPCR: quantitative PCR; RFLP: restriction fragment length polymorphism; SSU: small subunit of ribosomal RNA.

**Table 4 microorganisms-09-01101-t004:** Wild animals infected with *L.*
*mexicana*. Organs or tissues where the parasite was detected, as well as the techniques employed, are indicated.

Host	Prevalence	Organs/Tissue Analysed	Methods for Detection	Country	References
**Order Carnivora**					
*Conepatus chinga rex* (Molina’s hog-nosed skunk)	50%	Liver + skin + spleen (macerate)	Inoculation to hamster, isoenzyme analysis, hybridisation	Bolivia	[41]
*Urocyon* *cinereoargenteus* (fox)	100%	serum	ELISA	Mexico	[69]
**Order Chiroptera**					
*Pteronotus personatus*	25%	heart, liver, skin and spleen	PCR (kDNA), PCR (SSU)	Mexico	[70]
*Artibeus jamaicensis*	5.8%	heart, liver, skin and spleen	PCR (kDNA), PCR (SSU)	Mexico	[70]
*Artibeus lituratus*	7.3%	heart, liver, skin and spleen	PCR (kDNA), PCR (SSU)	Mexico	[70]
*Carollia sowelli*	4.4%	heart, liver, skin and spleen	PCR (kDNA), PCR (SSU)	Mexico	[70]
*Choeroniscus godmani*	23.1%	heart, liver, skin and spleen	PCR (kDNA), PCR (SSU)	Mexico	[70]
*Desmodus rotundus*	7.1%	heart, liver, skin and spleen	PCR (kDNA), PCR (SSU)	Mexico	[70]
*Dermanura phaeotis*	8.1%	heart, liver, skin and spleen	PCR (kDNA), PCR (SSU)	Mexico	[70]
*Glossophaga commissarissi*	75%	heart, liver, skin and spleen	PCR (kDNA), PCR (SSU)	Mexico	[70]
*Glossophaga soricina*	26.9%	heart, liver, skin and spleen	PCR (kDNA), PCR (SSU)	Mexico	[70]
*Leptonycteris curasoae*	50%	heart, liver, skin and spleen	PCR (kDNA), PCR (SSU)	Mexico	[70]
*Phyllostomus discolor*	100% (*n* = 1)	heart, liver, skin and spleen	PCR (kDNA), PCR (SSU)	Mexico	[70]
*Stumira lilium*	11.1%	heart, liver, skin and spleen	PCR (kDNA), PCR (SSU)	Mexico	[70]
*Stumira ludovici*	4%	heart, liver, skin and spleen	PCR (kDNA), PCR (SSU)	Mexico	[70]
**Order Didelphimorphia**					
*Marmosa mexicana* (Mexican mouse opossum)	66.7%	base of the tail	PCR (kDNA)	Mexico	[71]
**Order Pilosa**					
*Tamandua mexicana* (northern tamandua)	6.3%	spleen	PCR (ALAT), PCR (ITS1) + sequencing	Mexico	[79]
**Order Primates**					
*Alouatta palliate* (mantled howler monkey)	5%	serum	ELISA, IFAT and WB	Mexico	[80]
*Alouatta pigra* (Guatemalan black howler)	37.5%	serum	ELISA, IFAT and WB	Mexico	[80]
**Order Rodentia**					
*Heteromys gaumeri*	46.3%	base of the tail	PCR (kDNA)	Mexico	[71]
*Heteromys desmarestianus*	100%	base of the tail	PCR (kDNA)	Mexico	[71]
*Neotoma micropus* (woodrats)	7.3–50%	skin, ear tissue	Culture, PCR (kDNA), Culture of lesions + PCR + isoenzyme analysis of cultures	USA	[73,74]
*Neotoma floridana* (eastern woodrat)	100%	ear, foot	Smears, PCR	USA	[76]
*Handleyomys* (*Oryzomys) melanotis*	65–100%	skin (base-tail, lesions), liver and spleen	Culture, Mab, imprints and PCR (kDNA)	Mexico	[71,72,75,77]
*Ototylomis phyllotis*	75.5–100%	skin (base-tail), liver	Culture, Mab, imprints and PCR (kDNA)	Mexico	[71,72,75]
*Peromyscus attwateri*	100% (*n* = 1)	skin (neck)	PCR (ITS1) + sequencing	USA	[78]
*Peromyscus yucatanicus*	28.6–100%	skin (base-tail), heart and kidney, liver, spleen	Culture, PCR (kDNA) and imprints	Mexico	[71,72]
*Rattus rattus* (black rat)	2.9–19%	blood	Culture, PCR (kDNA) + RFLP/hybridisation	Venezuela, Brazil	[33,58]
*Reithrodontomys gracilis*	66.6%	skin (base-tail)	Culture, PCR (kDNA)	Mexico	[71]
*Sigmodon hispidus* (cotton rat)	58.8–100%	liver, skin (base-tail, lesion) and spleen	Imprints, culture, Mab and PCR (kDNA)	Mexico	[71,72,75,77]
*Trichomys apereoides*	27.8%	blood	PCR (kDNA) + hybridisation	Brazil	[58]

ALAT: alanine transaminase; ELISA: enzyme immune assay; IFAT: Immunofluorescence assay; ITS: internal transcriber spacer; kDNA: kinetoplast DNA; Mab: monoclonal antibodies; SSU: small subunit of ribosomal RNA.

**Table 5 microorganisms-09-01101-t005:** Wild animals infected with *L. infantum* (sin. *L. chagasi*) and *Leishmania* spp. in the Americas. Organs or tissues positive to the parasite, as well as techniques employed, are indicated.

Host	Prevalence	Organs/Tissue Analysed	Method of Detection	Country	Reference
**Order Carnivora**					
*Cerdocyon thous* (crab-eating fox)	4–100%	BM, heart, liver, lung, mesenteric LN, serum skin and spleen	Smears, ELISA, culture, PCR, inoculation to hamster, IFAT, xenodiagnosis vector, PCR (kDNA) + sequencing, PCR (kDNA) + sequencing *	Brazil	[81,82,83,84,85,86,87,89]
*Chrysocyon brachyurus* (maned wolf)	10–75%	BM, serum and skin	ELISA, IFAT, PCR (kDNA), PCR (kDNA) + sequencing, IC rk39, xenodiagnoses in vector	Brazil	[82,84,85,88,89,90]
*Eira barbara* (tayra)	n.s.	serum	DAT (*n* = 3)	Brazil	[92]
*Galictis cuja* (lesser grison)	n.s.	serum	DAT (*n* = 3)	Brazil	[92]
*Leopardus pardalis* (ocelot)	75%	serum	ELISA,	Brazil	[90]
*Lycalopex* (*Pseudalopex*) *vetulus* (hoary fox)	33.3%	BM, serum	IFAT, ELISA, PCR (kDNA)	Brazil	[84]
*Nasua nasua* (coati)	n.s.	serum	DAT (*n* = 2)	Brazil	[92]
*Speothos venaticus* (bush dogs)	33.3–100%	blood, LN, serum, skin, spleen and other tissues (liver, kidney, lung and large intestine)	PCR (kDNA), PCR (kDNA) + sequencing, histopathology, IHC, ELISA, IFAT, IC rk39, xenodiagnoses in vector	Brazil	[84,85,89,90,91]
*Panthera onca* (jaguar)	20–50%	blood, LN and serum	PCR (kDNA) + RFLP, ELISA, IC	Brazil	[90,93]
*Panthera tigris altaica* (Siberian tiger)	50%	serum	ELISA, IC	Brazil	[90]
*Panthera leo* (lion)	50–100%	blood, serum	PCR (kDNA) + RFLP, ELISA	Brazil	[90,94]
*Puma concolor* (cougar)	71.4%	blood, LN	PCR (kDNA) + RFLP	Brazil	[93]
*Procyon cancrivorus* (crab-eating racoon)	33.3%	kidney	PCR (kDNA) + sequencing *	Brazil	[86]
**Order Chiroptera**					
*Artibeus planirostris* (frugivorous)	7.4–16.7%	blood	PCR (kDNA), PCR (Cyt B) + sequencing	Brazil	[45,95]
*Artibeus lituratus* (frugivorous)	40.9%	blood	qPCR (kDNA)	Brazil	[95]
*Desmodus rotundus* (hematophagous)	50%	liver, skin	qPCR (kDNA), PCR (ITS1) + RFLP, PCR (kDNA), nPCR (SSU) + sequencing	Brazil	[14,44]
*Carollia perspicillata* (frugivorous)	3–27.3%	blood, spleen	Culture, qPCR (kDNA)*,* qPCR (SSU), PCR (kDNA), PCR (ITS2) + sequencing	Venezuela, Brazil and French Guiana	[95,96,97]
*Eumops perotis* (insectivorous)	11.1%	blood	PCR (kDNA), PCR (Cyt B) + sequencing	Brazil	[45]
*Eptesicus furinalis* (frugivorous)	100% (*n* = 1)	blood	PCR (kDNA), PCR (Cyt B) + sequencing	Brazil	[45]
*Glossophaga soricina* (nectarivorous)	0.7–100%	blood, liver and spleen	nPCR (SSU), PCR (kDNA), PCR (Cyt B) + sequencing, qPCR (kDNA), PCR (kDNA) and nPCR (SSU) + sequencing	Brazil	[44,45,66,95]
*Myotis nigricans* (insectivorous)	33.3%	liver	PCR (kDNA) + nPCR (SSU) + sequencing	Brazil	[44]
*Molossus molossus* (insectivorous)	0.5%–100%	blood, liver and spleen	nPCR (SSU), qPCR (kDNA), PCR (ITS1) + RFLP, PCR (kDNA), PCR (Cyt B) + sequencing, PCR (kDNA) + nPCR (SSU) + sequencing	Brazil	[14,44,45,66]
*Molossus pretiosus* (insectivorous)	21.1%	liver, skin	PCR (kDNA) + nPCR (SSU) + sequencing	Brazil	[44]
*Molossus rufus* (insectivorous)	20–100%	liver, spleen	qPCR (kDNA), PCR (ITS1) + RFLP, nPCR (SSU) + sequencing	Brazil	[14,44]
*Molossidae* spp. (insectivorous)	40%	liver, skin	PCR (kDNA)+ nPCR (SSU) + sequencing	Brazil	[44]
*Nyctinomops laticaudatus* (insectivorous)	40%	liver, skin	PCR (kDNA) and nPCR (SSU) + sequencing	Brazil	[44]
*Nyctinomops macrotis* (insectivorous)	60%	liver, skin	PCR (kDNA) + nPCR (SSU) + sequencing	Brazil	[44]
*Platyrrhynus lineatus* (frugivorous)	15.4%	blood	qPCR (kDNA)	Brazil	[95]
*Phyllostomus hastatus* (omnivorous)	5.9%	blood	PCR (kDNA), PCR (Cyt B) + sequencing	Brazil	[45]
*Phyllostomus discolor* (omnivorous)	100% (*n* = 1)	blood	qPCR (kDNA)	Brazil	[95]
*Pteronotus parnellii* (insectivorous)	100% (*n* = 1)	blood	PCR (SSU), PCR (GAPDP)	Brazil	[98]
*Bats* (n.s.)	0.1%	oral swab	PCR (SSU) + sequencing *	Brazil	[32]
**Order Cingulata**					
*Dasypus septemcinctus* (seven-banded armadillo)	100% (*n* = 1)	liver	PCR (kDNA) + sequencing *	Brazil	[86]
**Order Didelphimorphia**					
*Didelphis albiventris* (white-eared opossum)	6.3–22.2%	blood, BM, liver, lung, kidney, skin and spleen	Culture, PCR (ITS1) + RFLP, PCR (kDNA), PCR (kDNA) + sequencing, nPCR (SSU) + sequencing, PCR (kDNA), PCR (ITS1)	Brazil	[39,50,54,86,100]
*Didelphis aurita* (big-eared opossum)	6.3%	LN, serum and spleen	Spleen smears, PCR (kDNA) + hybridisation IC rk39	Brazil	[102]
*Didelphis marsupialis* (common opossum)	7.1–40.5%	blood, BM, liver, serum, skin and spleen	smears, Culture, inoculation to hamster + isoenzyme, Mab, PCR (kDNA) + hybridisation, IFAT, DAT, PCR+RFLP, nPCR (SSU), PCR (ITS1)	Brazil, Colombia and Venezuela	[38,103,104,105,106,107]
*Didelphis* sp. *D. albiventris* *D. aurita*	91.6%	blood, BM	PCR (kDNA) ELISA, FML-ELISA, smears, culture	Brazil	[101]
**Order Lagomorpha**					
*Lepus europaeus* (European hare)	n.s.	serum	DAT (n = 1)	Brazil	[92]
**Order Pilosa**					
*Myrmecophaga tridactyla* (giant anteater)	33.3%	heart, kidney, lung and mesenteric LN	PCR (kDNA) + sequencing *	Brazil	[86]
*Tamandua tetradactyla* (lesser anteater)	50–100%	BM, liver, lung and mesenteric LN	PCR (kDNA), PCR (ITS1) + sequencing	Brazil	[86,99]
**Order Primates**					
*Alouatta caraya* (black howler)	3.7%	ear tissue	PCR (ITS) + RFLP + sequencing	Brazil, Argentina	[64]
*Alouatta guariba* (brown howler monkey)	12.5	blood	PCR (kDNA)	Brazil	[108]
*Alouatta seniculus* (red howler monkey)	22.2%	blood	PCR (kDNA), PCR (ITS2), PCR (SSU), IC	French Guiana	[119]
*Aotus nigriceps* (black-headed night monkey)	100% (*n* = 1)	blood	qPCR (kDNA)	Brazil	[108]
*Callicebus nigrifons* (black-fronted titi)	33.3%	blood, liver, lung, intestine and spleen	qPCR (kDNA), IHC,	Brazil	[108]
*Callithrix jacchus* (white-tufted-ear marmoset)	100% (*n* = 1)	serum	DAT	Brazil	[92]
*Callithrix penicillata, C. jacchus*	26.9%	blood, skin	DAT, PCR + sequencing	Brazil	[109]
*Cebus xanthosternos* (golden-bellied capuchin)	60%	blood	qPCR (kDNA)	Brazil	[108]
*Leontopithecus chrysomelas* (golden-headed lion tamarin)	20%	blood	qPCR (kDNA)	Brazil	[108]
*Pithecia irrorata* (bald-faced saki)	50%	blood	qPCR (kDNA)	Brazil	[108]
*Saguinus imperator* (emperor tamarin)	100%	blood	qPCR (kDNA)	Brazil	[108]
**Order Rodentia**					
*Cavia aperea* (Brazilian guinea pig)	25%	heart	PCR (kDNA) + sequencing	Brazil	[86]
*Cerradomys (Oryzomys) subflavus*	25%	BM, liver and spleen	nPCR (SSU) + sequencing	Brazil	[50]
*Coendu (Sphiggurus) villosus* (prehensile tailed porcupine)	n.s.	serum	DAT (*n* = 2)	Brazil	[92]
*Coendou (Sphiggurus) spinosus* (Paraguayan hairy dwarf porcupine)	20%	heart, kidney, liver and spleen	PCR (kDNA) + sequencing	Brazil	[86]
*Clyomis laticeps*	5.2%	spleen	PCR (kDNA) + PCR (HSP70)	Brazil	[36]
*Dasyprocta azarae*	16.7%	spleen	PCR (kDNA) + PCR (HSP70)	Brazil	[36]
*Dasyprocta* sp.	n.s.	blood, skin	PCR (kDNA) + PCR (ITS), PCR (HSP70) + sequencing	Brazil	[110]
*Holochilus scieurus*	10%	skin, spleen	PCR (kDNA)	Brazil	[39]
*Hydrochoerus hydrochaeris* (capybara)	50%	lung	PCR (kDNA) + sequencing	Brazil	[86]
*Mus musculus* (house mice)	20%	BM, liver, tail–ear skin and spleen	nPCR (SSU) + sequencing	Brazil	[50]
*Nectomys squamipes*	7%	skin, spleen	PCR (kDNA)	Brazil	[39]
*Proechymis canicollis*	8.8%	skin, spleen	PCR + hybridisation	Colombia	[106]
*Proechymis cuvieri*	n.s.	blood, skin	PCR (kDNA) + PCR (ITS), PCR (HSP70) + sequencing	Brazil	[110]
*Rhipidomys mastacalis*	28.5%	liver	PCR (HSP70) + RFLP	Brazil	[21]
*Rattus norvegicus* (brown rat)	16.7%	liver, tail–ear skin,	nPCR (SSU) + sequencing	Brazil	[50]
*Rattus rattus* (black rat)	0.1–100%	blood, BM, liver, skin and spleen	PCR (kDNA), PCR (kDNA) + hybridisation, PCR (HSP70) + RFLP, PCR (kDNA), nPCR (SSU) + sequencing, PCR (HSP70) + RFLP	Venezuela Brazil	[21,39,50,53,58,107]
*Trichomys apereoides*	6.3–11.1%	skin, ear skin	PCR (kDNA) + hybridisation PCR (HSP70) + RFLP	Brazil	[21,58]
*Trichomys laurentis*	1%	spleen	PCR (kDNA)	Brazil	[36]
Wild animals infected with ***Leishmania* spp**. in the Americas					
**Host**	**Prevalence**	**Organs/tissue Analysed**	**Method of Detection**	**Country**	**Reference**
**Order Carnivora**					
*Canis latrans* (coyote)	1.6%	serum	IC rAgK39	USA	[112]
*Cerdocyon thous* (crab-eating fox)	15.3–100%	blood, serum and skin	qPCR (kDNA), IFAT, IC	Brazil	[114,115]
*Chrysocyon brachyurus* (maned wolf)	42.9%	blood	qPCR (kDNA)	Brazil	[114]
*Lontra longicaudis* (neotropical otter)	50%	blood	qPCR (kDNA)	Brazil	[114]
*Lycalopex (Pseudalopex) griseus* (South American grey fox)	37.5%	blood	PCR (kDNA) + sequencing	Argentina	[113]
*Lycalopex (Pseudalopex) vetulus* (hoary fox)	7.1–50%	blood, serum	qPCR (kDNA), IFAT	Brazil	[114,115]
*Nasua nasua* (coati)	50%	blood	qPCR (kDNA), IFAT	Brazil	[114,116]
*Puma concolor* (cougar)	100% (*n* = 1)	blood	qPCR (kDNA)	Brazil	[114]
*Spheotos venaticus* (bush dog)	33.3–100%	Blood, serum, liver and LN	ELISA, PCR (kDNA)	Brazil	[91,117]
*Vulpes fulvus* (American red fox)	9.1%	serum	IC rK39	USA	[112]
*Urocyon cinereoargenteus* (gray fox)	2%	serum	IC rK39	USA	[111]
**Order Chiroptera**					
*Molossus molossus* (insectivorous)	7.4%	liver	PCR (kDNA)+ nPCR (SSU) + sequencing	Brazil	[44]
*Molossus pretiosus* (insectivorous)	5.2%	liver	PCR (kDNA)+ nPCR (SSU) + sequencing	Brazil	[44]
*Nyctinomops macrotis* (insectivorous)	6.7%	liver	PCR (kDNA)+ nPCR (SSU) + sequencing	Brazil	[44]
**Order Pilosa**					
*Myrmecophaga tridactyla* (giant anteater)	36.4%	blood	qPCR (kDNA)	Brazil	[114]
*Tamandua tetradactyla* (lesser anteater)	33.3%	blood	qPCR (kDNA)	Brazil	[114]
**Order Primates**					
*Alouatta guariba* (brown howler monkey)	37.5%	blood	PCR (kDNA)	Brazil	[91]
*Aotus nigriceps* (black-headed night monkey)	20%	blood	qPCR (kDNA)	Brazil	[114]
*Chiropotes satanas* (black-bearded saki)	50%	blood	qPCR (kDNA)	Brazil	[114]
*Lagothrix cana* (gray-woolly monkey)	33.3%	blood	qPCR (kDNA)	Brazil	[114]
*Leontopithecus chrysomelas* (golden-headed lion tamarin)	16.7%	blood	qPCR (kDNA)	Brazil	[114]
**Order Rodentia**					
*Rattus rattus* (black rat)	9.1%	serum	IFAT	Dominican Republic	[118]
*Sciurus granatensis* (red-tailed squirrel)	100% (*n* = 1)	blood	nPCR (SSU)	Venezuela	[103]

BM: bone marrow; Cyt B: cytochrome B; DAT: direct agglutination test; FML: fucose-mannose ligand; GADPH: glyceraldehyde phosphate dehydrogenase; HSP70: heat shock protein 70kDa; IHC: immunohistochemistry; IC: immunochromatography; ELISA: enzyme immune assay; IFAT: immunofluorescence assay; ITS: internal transcriber spacer; kDNA: kinetoplast DNA; LN: lymph node; nPCR: nested PCR; n.s.: not specified; qPCR: quantitative PCR; RFLP: restriction fragment length polymorphism; SSU: small subunit of ribosomal RNA. *** Probably *L. infantum,* according to the sequence.

**Table 6 microorganisms-09-01101-t006:** Wild animals reported to be positive for *L. infantum* from Europe, Asia and Africa. Organs or tissues where the parasite was detected are indicated, as well as the techniques employed for detection.

Host	Prevalence	Organs/Tissue Analysed	Methods for Detection	Country	References
**Order Carnivora**					
*Canis aureus* (golden jackal)	3–11.6%	blood, BM, liver, LN, serum, spleen	qPCR (ITS1), PCR (kDNA), IC rk39, smear, culture, PCR (α-tubulin and GAPDH)	Georgia, Israel, Iran and Romania	[120,121,122,123]
*Canis lupus* (grey wolf)	6–100%	blood, hair, liver, LN, skin, serum, spleen	PCR (cysteine protease B), qPCR (kDNA), PCR (kDNA) + RFLP, PCR (ITS2) + RFLP, PCR (kDNA) + sequencing, ELISA	Croatia, Italy, Spain	[124,125,126,127,128,129,130,131,132]
*Felis silvestris* (wildcat)	25–100%	liver, LN, skin, spleen	qPCR (kDNA), PCR (ITS2) + sequencing, PCR (kDNA) + sequencing, qPCR (kDNA) + RFLP + sequencing	Spain	[127,128,133]
*Genetta genetta* (common genet)	10–100%	blood, liver, skin and spleen	PCR (kDNA) + RFLP, qPCR (kDNA), PCR (ITS2) + sequencing, PCR (kDNA) + sequencing, qPCR (kDNA) + RFLP + sequencing, PCR (kDNA & ITS2) + RFLP	Spain	[125,127,128,129,133,134]
*Herpestes ichneumon* (Egyptian mongoose)	4.7–28.6%	blood, spleen,	PCR (kDNA) + RFLP, PCR (kDNA) + sequencing, PCR (ITS1)	Spain, Portugal	[125,135]
*Lutra lutra* (Eurasian otter)	70%	spleen	PCR (kDNA) + sequencing	Spain	[127]
*Lynx pardinus* (Iberian lynx)	25%	blood, spleen,	PCR (kDNA) + RFLP	Spain	[125]
*Martes foina* (beech marten)	29–100%	liver, LN, hair, skin and spleen	qPCR (kDNA), qPCR (ITS2) + sequencing, PCR (kDNA) + sequencing, qPCR (kDNA) + sequencing, PCR (kDNA & ITS2) + RFLP	Spain	[127,128,129,131,133,137]
*Martes martes* (European pine marten)	30–62%	blood, liver, spleen	PCR (kDNA) + RFLP, qPCR (kDNA), qPCR (ITS2) + sequencing, PCR (kDNA) + sequencing	Spain	[127,133,134]
*Meles meles* (European badger)	26–53%	liver, spleen	qPCR (kDNA), PCR (ITS2) + sequencing, PCR (kDNA) + sequencing	Spain, Italy	[132,133]
*Mustela lutreola* (European Mink)	2.1–50%	liver, spleen, serum	qPCR (kDNA), PCR (ITS2) + sequencing, PCR (ITS1), ELISA	Greece, Spain	[133,136]
*Mustela putorius* (European polecat)	25%	liver, spleen	qPCR (kDNA), PCR (ITS2) + sequencing	Spain	[133]
*Mustela vison* (American mink)	100% (*n* = 1)	liver, spleen	qPCR (kDNA)	Spain	[137]
*Panthera tigris* (Tiger)	25%	serum, LN and swab (oral, conjunctival and nasal)	IFAT, qPCR	Italy	[138]
*Sciurus vulgaris* (red squirrel)	20%	liver, skin, pleen	qPCR (kDNA)	Spain	[137]
*Ursus arctos* (brown bear)	100% (*n* = 1)	liver, skin, spleen	PCR (kDNA), PCR (ITS2) + RFLP	Spain	[129]
*Vulpes vulpes* (red fox)	2.6–74.6%	blood, BM, hair, liver, LN, skin, spleen, serum	PCR (Repeat Region), PCR (kDNA), PCR (kDNA) + RFLP, qPCR (kDNA), qPCR (ITS2) + sequencing, qPCR (ITS1) + RFLP, PCR (α-tubulin and GAPDH) + sequencing, PCR (kDNA) + RFLP, PCR (kDNA) + sequencing, PCR (ITS2) + RFLP, PCR (ITS1) + sequencing, ELISA, IFAT, WB, IC rk39, smear, culture	France, Georgia and Greece, Iran, Italy and Spain	[120,122,125,127,128,129,131,132,139,140,141,142,143,144,145,146]
**Order Chiroptera**					
*Pipistrellus pipistrellus* (common urban bat)	59.2%	blood clot, hair, spleen	PCR (Repeat region) + sequencing	Spain	[149]
Order Diprotodontia					
*Macropus rufogriseus* (Bennett’s wallaby)	33.3%	blood, BM, liver, lung, LN, kidney, skin, spleen	PCR (ITS1 and ITS2) + sequencing, IC rk39	Spain	[147]
**Order Eulipotyphla**					
*Atelerix algirus* (Algerian hedgehog)	100%	blood, eye swab, heart, kidney, liver, LN, skin, spleen	PCR (kDNA), PCR (ITS1), PCR (mini-exon), PCR (Repeat region), PCR (SSU), smear	Tunisia	[150,151]
*Erinaceus europaeus* (European hedgehog)	34.4–100%	hair, serum, skin, spleen	qPCR (kDNA), ELISA	Spain	[131,137]
**Order Lagomorpha**					
*Lepus europaeus* (European hare)	0.9–43.6%	blood, spleen, serum	PCR (kDNA) + RFLP, PCR (ITS1), PCR (ITS1) + sequencing, ELISA, IFAT	Greece, Italy and Spain	[136,152,155,156]
*Lepus granatensis* (Iberian hare)	10.1–100%	hair, skin, spleen, serum	PCR (kDNA) + RFLP, qPCR (kDNA), nPCR (SSU), IFAT, DFA	Spain	[152,153,154]
*Oryctolagus cuniculus* (European rabbit)	0.6–59%	blood, BM, hair, heart, liver, LN, skin, spleen, serum	qPCR (kDNA), PCR (ITS1) + RFLP, ELISA, nPCR (SSU), qPCR (kDNA) + RFLP + sequencing, PCR (kDNA) + RFLP, PCR (ITS2) + RFLP, PCR (ITS1), smears, culture, IFAT, DFA, ELISA, IC rk39	Greece, Italy and Spain	[128,129,136,145,153,154,157,158]
**Order Primates**					
*Pongo pygmaeus* (north west Bornean orangutan)	100%	BM, serum	Microscopy, IFAT, nPCR (ITS1)	Spain	[148]
**Order Rodentia**					
*Apodemus sylvaticus* (wood mouse)	20–50%	blood, BM, liver, skin, spleen	PCR (ITS1) + sequencing, PCR-ELISA (kDNA), qPCR (kDNA) + RFLP + sequencing, PCR (ITS2) + RFLP, smear, culture	Spain	[128,129,159]
*Crocidura russula* (white-toothed shrew)	13.3%	blood and/or spleen	qPCR (kDNA)	Spain	[160]
*Mus musculus* (house mouse)	22–50%	blood, BM, liver, skin, spleen	qPCR (kDNA) + sequencing, PCR (ITS1) + sequencing, PCR-ELISA (kDNA), nPCR (SSU and ITS1) + sequencing, smear	Morocco, Portugal and Spain	[162,163]
*Mus spretus* (Algerian mouse)	4.3–42.9%	blood, liver, skin, spleen and serum	qPCR (kDNA), ELISA	Spain	[137,160]
*Nesokia indica* (short-tailed bandicoot rat)	39%	liver, skin, spleen,	nPCR (kDNA), smear	Iran	[162]
*Rattus norvegicus* (brown rat)	5.9–100%	hair, liver, skin, spleen	nPCR (SSU), nPCR (ITS1) + sequencing, qPCR (kDNA), PCR (kDNA), PCR (kDNA) + RFLP, PCR (ITS2) + RFLP, smear	Greece, Morocco, Portugal and Spain	[129,131,162,163,166]
*Rattus rattus* (black rat)	7.5–33.3%	blood, BM, liver, skin, spleen	PCR (kDNA) + sequencing, PCR (ITS1) + sequencing, PCR-ELISA (kDNA), nPCR (SSU), nPCR (ITS1) + sequencing, smear, culture, inoculation to hamster, isoenzymes	Italy, Morocco, Saudi Arabia and Spain	[159,163,164,165]

BM: bone marrow; Cyt b: cytochrome B; DFA: direct fluorescence antibody assay; GADPH: glyceraldehyde phosphate dehydrogenase; IC: immunochromatography; ELISA: enzyme immune assay; IFAT: immunofluorescence assay; ITS: internal transcriber spacer; kDNA: kinetoplast DNA; LN: lymph node; nPCR: nested PCR; qPCR: quantitative PCR; RFLP: restriction fragment length polymorphism; SSU: small subunit of ribosomal RNA; WB: Western blot.

**Table 7 microorganisms-09-01101-t007:** Wild animals reported positive for *L. major*. Organs or tissues where the parasite was detected, and the techniques employed are indicated.

Host	Prevalence	Organs/Tissue Analysed	Methods for Detection	Country	References
**Order Chiroptera**					
*Nycteris hispida*	100% (*n* = 1)	spleen	qPCR (kDNA and ITS1) + sequencing	Ethiopia	[172]
**Order Eulipotyphla**					
*Atelerix algirus* (Algerian hedgehog)	36.8–100%	blood, eye swab, heart, kidney, liver, LN, skin, spleen	qPCR (kDNA), PCR (kDNA), PCR (ITS1) + RFLP, nPCR (kDNA), PCR (ITS1) + sequencing + RFLP, PCR (mini-exon) + sequencing + RFLP, nPCR (Repeat region) + sequencing + RFLP, PCR (SSU) + sequencing, smear, ELISA, WB	Algeria, Tunisia	[150,151,175]
*Hemiechinus auritus* (long-eared hedgehogs)	33.3–53.3%	liver, skin, spleen	nPCR (ITS1) + sequencing, nPCR (kDNA), semi-nPCR (kDNA), smear	Iran	[173,174]
*Paraechinus aethiopicus* (desert hedgehog)	40–100%	blood, eye swab, kidney, liver, LN, skin, spleen, serum	qPCR (kDNA), PCR (ITS1) + RFLP, PCR (kDNA and SSU) + sequencing, nPCR (Repeat region) + RFLP + sequencing, ELISA, WB	Algeria, Tunisia	[151,175]
**Order Primates**					
*Cercopithecus mitis* (syke’s monkeys)	67.2%	serum	ELISA, WB	Kenya	[176]
*Chlorocebus aethiops* (vervet monkeys)	60.6%	serum	ELISA, WB, lymphocyte proliferation assay	Kenya	[176]
*Gorilla gorilla* (gorilla)	13.2%	faeces	qPCR (SSU), qPCR (SSU) + sequencing, PCR (ITS) + sequencing, PCR (Cytb) + sequencing	Cameroon	[15]
*Papio cynocephalus anubis* (olive baboons)	77.2%	serum	ELISA, WB	Kenya	[176]
**Order Rodentia**					
*Gerbillus nanus*	11.8%	liver, skin, spleen	PCR (kDNA), smear	Iran	[171]
*Meriones hurrianae*	7.7%	liver, skin, spleen	PCR (kDNA), smear	Iran	[171]
*Meriones libycus*	5.7–100%	liver, skin, spleen	PCR (kDNA), nPCR (ITS1), PCR (ITS1) + RFLP + sequencing, semi-nPCR (kDNA), PCR (Cytb) + sequencing, nPCR (ITS1) + sequencing, nPCR (ITS2) + RFLP, smear, inoculation to hamster, inoculation to BALB/c mice	Iran	[167,168,169,170,173,177,178]
*Meriones persicus*	33%	skin	PCR (ITS1) + RFLP + sequencing, smear	Iran	[168]
*Meriones tristrami*	58.3%	skin	PCR (ITS1) + RFLP	Israel	[179]
*Microtus guentheri*	16.5%	skin	PCR (ITS1) + RFLP	Israel	[179]
*Microtus socialis*	50%	liver, skin, spleen	smear	Iran	[161]
*Mus musculus* (house mouse)	2.3–33%	liver, skin, spleen	PCR (ITS1) + RFLP, smear	Israel, Morocco and Iran	[161,179]
*Nesokia indica*	8–63.4%	liver, skin, spleen	PCR (ITS1) + RFLP + sequencing, PCR (kDNA), nPCR (kDNA), smear	Iran	[161,167,168]
*Rhombomys opimus* (great gerbil)	13.4–35%	skin	PCR (ITS1) + RFLP + sequencing, semi-nPCR (kDNA), PCR (Cytb) + sequencing, smear, IHC, inoculation to hamster, inoculation to BALB/c mice	Iran	[168,169]
*Tatera indica*	3.7–50%	liver, skin, spleen	PCR (kDNA), PCR (ITS1) + RFLP + sequencing, semi-nPCR (kDNA) + sequencing, PCR (Cytb), smear	Iran	[168,169,171]

Cytb: cytochrome B; IHC: immunohistochemistry; ELISA: enzyme immune assay; ITS: internal transcriber spacer; kDNA: kinetoplast DNA; LN: lymph node; nPCR: nested PCR; qPCR: quantitative PCR; RFLP: restriction fragment length polymorphism; SSU: small subunit of ribosomal RNA; WB: Western blot.

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
