# Peer review of "A Systematic Review (1990–2021) of Wild Animals Infected with Zoonotic Leishmania"

_microorganisms, 2021, doi:10.3390/microorganisms9051101_

Round 1
Reviewer 1 Report
Congratulations on the extensive and thorough review; however, I would like you to clarify some issues.
Line 33-34: It said "Most people and animals infected by the parasite do not develop symptoms but, if present, the disease can follow three basic clinical forms: cutaneous, mucocutaneous and visceral". However, mucocutaneous lesions are only observed in human beings; therefore I suggest modifying the sentence
Table I. "Man" is not a correct form to speak about "Human beings"
Table I. To Include the meaning of CL, DCL, MCL, PKDL, ...
Line 134. The epidemiological role of "equines" is not yet known. You can detect DNA or antibodies against L. infantum but it is not known if they play a relevant role.
Line 135. It said "However, dogs were found with similar or even lower prevalence than wildlife in some 135 outbreaks". Have you taken into account that the dogs used repellents or dogs could be vaccinated (preventive measures) in the case of some outbreaks as in Madrid? In that case, the prevalence in owner dogs should be lower, don't you think?
Line 183: To review "CD$*T"
Line 228: To change "of Leihsmania" by "of Leishmania"
Line 535: To change "(Alcover et al., 2020) by (138).
Line 593: It said "Surprinsingly, one study found DNA of L major in the feces of Gorillas (Gorilla gorilla) in Cameroon and the authors pointed out to the ingestion of phlebotomines by the animals". This statement should be discussed, I don't consider correct to include it in this review. ADN could also have been detected in feces from rabbit / hare burrows where the life cycle of sandflies is completed. Explain this sentence.
References: To include:
- Cantos-Barrera et al., 2020. Clinical leishmaniosis in a captive Eurasian otter (Lutra lutra) in Spain: a case report
- Cavalera et al., 2020. Clinical, haematological and biochemical findings in tigers infected by Leishmania infantum
- Miró et al., 2018. First report of L. infantum infection in the endangered orangutan (Pongo pygmaeus pygmaeus) in Madrid, Spain.
Author Response
Please, see the attachment

Reviewer 2 Report
The manuscript requires gramatical corrections, some of which are indicated in the minor comments, but not by all means all. For example, it was difficult to identify in the Results section what was found in this review or was previous knowledge.
Major issues:
- In the taxonomic revision, the authors fail to include the reference for the article where sub-genus Mundinia is created. Espinosa OA, Serrano MG, Camargo EP, Teixeira MMG, Shaw JJ. An appraisal of the taxonomy and nomenclature of trypanosomatids presently classified as Leishmania and Endotrypanum. Parasitology. 2018 Apr;145(4):430-442. doi: 10.1017/S0031182016002092. https://pubmed.ncbi.nlm.nih.gov/27976601/ Instead, they use a 20 year old reference by Cupolillo, even though, there are much more recent revisions, some of which are included in the references (5-7), but these are hardly mentioned as significant, despite being much more recent and taking into account more recent molecular data and isolates. In any case, this type of review is hardly the place for a discussion of Leishmania taxonomy, and the authors should simply present the most recent or most consensual list of species, in my opinion.
- There is no reason for the Introduction to be so extensive and divided into sections, when the main focus of the review is zoonotic leishmaniasis, more specifically wild animals. As such, the Introduction should be reduced and sumarized for the benefit of any readers not familiar with Leishmania. Table 1 should also be reduced and included instead in the Materials and Methods.
- Why exclusion of “studies with negative results to Leishmania infections”? While it is understandable that the authors want to include species with positive results, I would expect that negative results in relation to positive results will also provide an indication of how likely it is, or not, that a given species is a potential reservoirt.
- The Results section reads in parts as an Introduction, and much content does not seem directly related to non-domestic animals findings. It should describe findings from the search, as per the objectives. Attention should be paid to the verb tenses used.
- Tables: The Species should be in a separate column for clarity.
- The authors should distinguish clearly between parasite, DNA or serological evidence for infection.
- L. donovani and L. tropica should be treated in separate sections, even if short.
- In the conclusions, the authors state that L. infantum is the most widely distributed species in terms of range of species that it infects, but it could be a consequence of more exaustive investigations. As the authors did not include negative results, it is difficult to ascertain if Leishmania species are not found because they are not searched for or because they do not establish infection.
Minor issues:
- Throughout, “Man” should be replaced with “Humans”.
- The authors use both leishmaniosis and leishmaniasis in the manuscript. It may be that they use leishmaniosis to refer to the zoonotic disease and leishmaniasis when referring to human disease, but, if so, this should be indicated in the Introduction. If not, they should choose one term for consistency.
- Figure 1 may be confusing to those not familiar with the Leishmania life cycle, as the sand fly appears near the vertebrate hosts, and then a promastigote appears to emerge from the vertebrate host. The authors should also be careful because sand flies do not feed directly on blood, but from a pool of blood made in the skin. (for quick reference see: https://www.ecdc.europa.eu/en/disease-vectors/facts/phlebotomine-sand-flies and associated references) As such, parasites present in the skin rather than blood may be taken up. And the presence of parasites in healthy skin is usually taken as necessary for transmission. The Figure is not necessary at all for this review, though.
Grammar issues, examples:
- Line 35: Leishmanisais to Leishmaniasis
- Line 35: vectorborne to vector-borne
- Line 43: collaborate with the WHO
- Line 44: updated instead of actualized
- Line 81: Genera not genuses
Reviewer 3 Report
This article is an extensive and highly detailed review of the wildlife species that have been found infected or exposed to Leishmania spp. The authors have put great effort and they gathered a large amount of information that is clearly presented in an understandable way.
General comments:
- The authors should keep in mind that the size of the manuscript is quite large and sometimes it is difficult to follow. For this reason, they should consider limiting some parts of the article that do not directly serve its purpose. In particular, the introduction section is very extensive and offers very basic information about the protozoan which is not relevant to the scope of this manuscript. Thus, it does not really add something important to the background that should be acquired by a reader who is interested in this particular topic.
For example:
The subsection “1.2 Life cycle” could be avoided or at least be limited. Especially, Lines 97-120 are irrelevant to the topic and create confusion for the reader.
Lines 129-152: The title of this subsection “1.3 Control of outbreaks and type of hosts” should be changed. The authors do not cover adequately the first part concerning the control of outbreaks and they could delete it as it is not relevant to the topic. They should also consider being more specific about the kind of reservoirs that the different wild animals may serve as. For example, in this section, the authors could mention the wild animal species that are deemed as secondary reservoirs till now. A reference to the species that has been shown to be able to transmit the parasite to phlebotomus species through xenodiagnosis studies, would certainly add value to this article and would highlight the importance of some wildlife species compared to others, for which only data concerning the presence of infection or exposure to the parasite exist.
- The authors should consider discussing the role of certain animal species (e.g. the most studied species such as carnivores and rodents, the species living in close proximity to humans, the species presenting higher prevalence of Leishmania infection, etc) in the epidemiology of Leishmaniasis in the different sections of the manuscript. For example, they could discuss the role of crab-eating fox in Lines 380-382 or the role of red fox in Lines 514-523, which are widespread species that are found in different ecosystems, from forests to residential areas, and can be involved in the transmission cycle of the protozoan to humans and wild and domestic animals through the different Phlebotomus species. This addition should be considered by the authors, at least for species that are of increasing importance for public health. Besides, the title of the manuscript mentions the zoonotic aspect.
- The authors could refer, throughout the manuscript, to the clinical signs that have been reported in the investigated wild animal species following Leishmania infection like they have already done in Lines 358-361 (e.g. add some more information in Lines 393-394, 522-523 and elsewhere throughout the manuscript).
- The authors should handle the reference to serological results with caution. There is a tension throughout the manuscript to refer to seropositivity as an indication or proof of Leishmania infection (e.g. Lines 230-232 and elsewhere in the manuscript) or presenting seroprevalence and Leishmania DNA prevalence in a comparable way (e.g. Lines 240-243). The authors should make clear that serological examination is used to investigate the exposure of animals to Leishmania parasites and avoid mentioning that serology provides evidence of Leishmania infection. It is preferable to report the seroprevalence and Leishmania DNA prevalence without comparing them in terms of sensitivity. Besides, as far as it concerns the serological methods, the cut-off values are not well established or validated in wild animal species. So, the seroprevalence detected in the different studies should be interpreted with caution as it may be over or underestimated. It is also important to mention that the kinetics of IgG antibodies against Leishmania parasites is still not well understood and studied in wild animals and presents great differences among the different animal species. The authors could also provide some details on the serological methods used in wild animals (e.g. IFAT, DAT) such as the serological titers reported in some cases.
- Some sections, like section “3.5. Wild animals infected with Leishmania infantum (L. chagasi) in the Americas” are difficult to follow. A lot of information that is already presented in Table 5 including the tissues examined or the PCR protocols used, is repeated in this section. The authors should consider re-organizing this section so as to be legible and provide valuable information on the infected species like in Lines 404-411 where they greatly summarize data on bats, while also mentioning the importance of their feeding habits and ecological niche. For example, Lines 413-428 and 430-469 should be reconsidered to provide information other than the already provided in the Table. The authors should re-check the other sections as well so as to avoid just repeating the already presented information.
- The authors repeat several times that kDNA PCR is more sensitive than other PCR targets (e.g. Lines 239, 254, 342, 585, 608) even in the same section. Maybe they could make a summary of the most common PCR targets used in wild animal samples and their sensitivity, the most common biological samples together with the possibility to detect Leishmania DNA with molecular techniques. Such an effort is already made by the authors in Section “3.2. Wild animals infected with zoonotic Leishmania (Viannia) spp.”. Maybe they could generalize this approach to cover the sections referring to the other Leishmania species as well.
- There are several grammar mistakes throughout the manuscript that should be corrected by the authors. I indicatively mention the following:
Line 244: change “zonotic” to “zoonotic”
Line 275: change “sixeen” to “sixteen”
Line 298: “Leishmania amazonensis” should be in italics
Line 303: change “analized” to “analyzed”
Line 309: delete the “,” at the end of the sentence
Line 339: rephrase to “samples from seven gray foxes…”
Line 356: delete the “of”
Line 388: change “freee” to “free”
Line 391: change “demonstrated” to “demonstrate”
Line 431: change “sequending” to “sequencing”
Line 432: change “especies” to “species”
Line 460: change “leves” to “level”
Line 625: change “developed” to “conducted”
Line 653: “Leishmania” in italics
Specific comments:
Lines 388-389: the sentence should be rephrased to be clear.
Lines 533-535: The authors should check the information they provide in this sentence. The reference number 133 concerns the detection of L.infantum DNA in European hedgehog through qPCR, not ELISA. Moreover, the reference Alcover et al., 2020 should be replaced with a numbered citation.
Table 3: the last references should be numbered.
Line 363: The reference (McHugh et al., 2003) should be numbered.
Line 560: The reference (Ibrahim et al., 1992) should be numbered
Lines 644 and 645: The reference (Morsy et al., 1999) should be numbered
Round 2
Reviewer 1 Report
I agree with the changes and suggestions
Reviewer 2 Report
I am happy with the changes done to the manuscript. The tables are much more informative, and the comments about sampling and detection methods make this review much more useful for the reader.
In answer to the authors, negative results could have been presented as a summary on a separate table as Supplementary Material, or mentioned as a means to ascertain the sampling effort, but I don't consider it a requirement for acceptance. Indeed the review is ambitious and already quite long.
There is only one minor issue that I have spotted and that would improve the quality of the writing, which is the repeated use of "like" when the most appropriate expression would be "such as", prior to giving an example, rather than a likeness.
Reviewer 3 Report
The manuscript is highly improved following the changes made by the authors. The authors made all the suggested changes and added value to this manuscript. Some minor spelling/grammar mistakes still exist.